# Translation of 5′ leaders is pervasive in genes resistant to eIF2 repression

Dmitry E Andreev[1,2]*[†], Patrick BF O'Connor[2][†], Ciara Fahey[3], Elaine M Kenny[3], Ilya M Terenin[1], Sergey E Dmitriev[1], Paul Cormican[3], Derek W Morris[3], Ivan N Shatsky[1]*, Pavel V Baranov[2]*

[1]Belozersky Institute of Physico-Chemical Biology, Lomonosov Moscow State University, Moscow, Russia; [2]School of Biochemistry and Cell Biology, University College Cork, Cork, Ireland; [3]Department of Psychiatry and Institute of Molecular Medicine, Trinity College Dublin, Dublin, Ireland

**Abstract** Eukaryotic cells rapidly reduce protein synthesis in response to various stress conditions. This can be achieved by the phosphorylation-mediated inactivation of a key translation initiation factor, eukaryotic initiation factor 2 (eIF2). However, the persistent translation of certain mRNAs is required for deployment of an adequate stress response. We carried out ribosome profiling of cultured human cells under conditions of severe stress induced with sodium arsenite. Although this led to a 5.4-fold general translational repression, the protein coding open reading frames (ORFs) of certain individual mRNAs exhibited resistance to the inhibition. Nearly all resistant transcripts possess at least one efficiently translated upstream open reading frame (uORF) that represses translation of the main coding ORF under normal conditions. Site-specific mutagenesis of two identified stress resistant mRNAs (PPP1R15B and IFRD1) demonstrated that a single uORF is sufficient for eIF2-mediated translation control in both cases. Phylogenetic analysis suggests that at least two regulatory uORFs (namely, in SLC35A4 and MIEF1) encode functional protein products.

*For correspondence:
cycloheximide@yandex.ru (DEA);
shatsky@genebee.msu.su (INS);
brave.oval.pan@gmail.com (PVB)

[†]These authors contributed equally to this work

Competing interests: The authors declare that no competing interests exist.

## Introduction

Protein synthesis, as one of the most energy consuming processes in the cell, is under stringent regulation. In eukaryotes, the activity of many components of the translational machinery is modulated by various post-translational modifications in order to adjust either global or mRNA-specific translation. One of the better studied cases of translational control is the phosphorylation of eukaryotic initiation factor 2 (eIF2) (*Sonenberg and Hinnebusch, 2009*).

eIF2 forms the ternary complex (TC) with GTP and Met-tRNAi and is loaded onto the 40S ribosome to enable it to recognize a start codon, after which eIF2*GDP is released. GDP is then recycled to GTP by guanine exchange factor (GEF), eIF2B, to enable another round of initiation. During various stress conditions the cell triggers the integrated stress response (ISR) by activating any of four kinases, EIF2AK1 (also known as [a.k.a.] HRI), EIF2AK2 (a.k.a. PKR), EIF2AK3 (a.k.a. PERK), or EIF2AK4 (a.k.a. GCN2), that phosphorylate the alpha subunit of eIF2 at Ser51 (*Baird and Wek, 2012*). Instead of a rapid recycling, eIF2B forms a stable complex with phosphorylated eIF2. The concentration of eIF2 is higher than that of eIF2B, therefore even phosphorylation of a modest number of eIF2 molecules rapidly reduces the pool of active eIF2B resulting in the general inhibition of total protein synthesis (*Hinnebusch, 2014*).

While the general suppression of translation conserves cellular resources, the active synthesis of certain factors is required to respond to the consequences of stress. Mammalian genes whose expression is known to evade translational arrest triggered by eIF2 phosphorylation include *ATF4* (*Lu et al., 2004*; *Vattem and Wek, 2004*), *PPP1R15A* (a.k.a. *GADD34*) (*Lee et al., 2009*), *ATF5*

**eLife digest** Proteins carry out essential tasks for living cells and genes contain the instructions to make proteins within their DNA. These instructions are copied to make a molecule of mRNA, and a molecular machine known as a ribosome then reads and translates the mRNA to build the protein.

The first step in the translation process is called 'initiation' and requires a protein called eIF2 to work together with the ribosome. This step involves identifying an instruction called the start codon that marks the beginning of the mRNA's coding sequence. The section of an mRNA molecule before the start codon is not normally translated by the ribosome and is hence called the 5′ untranslated region.

Building proteins requires energy and resources, and so it is carefully regulated. If a cell is stressed, such as by being exposed to harmful chemicals, it makes fewer proteins in order to conserve its resources. This down-regulation of protein production is achieved in part by the cell chemically modifying its eIF2 proteins to make them less able to initiate translation. However, stressed cells still continue to make more of certain proteins that help them to combat stress. The mRNA molecules for some of these proteins contain at least one other start codon in the 5′ untranslated region. The sequence that would be translated from such a start codon is known as an upstream open reading frame (or uORF for short)—and this feature is thought to help certain proteins to still be expressed despite low levels of active eIF2. Andreev, O'Connor et al. have now analysed which mRNAs are translated in human cells that have been treated with a chemical that induces stress and makes the eIF2 protein less able to initiate translation. To do so, a technique called ribosome profiling was used to identify all of the mRNA molecules bound to ribosomes shortly after treatment with this chemical.

Overall translation of most mRNAs in stressed cells was reduced to a quarter of the normal level. However, Andreev, O'Connor et al. observed that the translation of a few mRNAs continued almost as normal, or even increased, after the chemical treatment. Notably, most of these mRNAs encoded regulatory proteins, which are not required in large amounts. With one exception, all of these resistant mRNAs contained uORFs. In unstressed cells, these uORFs were efficiently translated, while the same mRNA's coding sequences were translated less efficiently. Andreev, O'Connor et al. suggest that these two features could be used to identify mRNAs that are still translated into working proteins when cells are stressed. Further work is now needed to explore the mechanisms by which translation of these uORFs allows mRNAs to resist the stress.

(*Watatani et al., 2008*; *Zhou et al., 2008*; *Hatano et al., 2013*), and *DDIT3* (a.k.a. *CHOP*) (*Jousse et al., 2001*; *Chen et al., 2010*).

*ATF4* and *ATF5* are believed to be regulated through the mechanism known as delayed reinitiation, initially characterized for the yeast *GCN4* (a functional analogue of *ATF4*) (*Hinnebusch, 1997*). This requires at least two upstream open reading frames (uORFs). In ATF4 mRNA, after translation termination at the first uORF, the 40S resumes scanning albeit without the TC. The distance scanned by this ribosome subunit before it reacquires the TC depends on TC availability. Under normal conditions most of the 40S is quickly reloaded with TC and therefore can reinitiate at the second uORF. Under stress conditions (i.e., low eIF2 availability), a larger fraction of 40S subunits scan past the second uORF initiation codon before binding of the TC, thereby enabling reinitiation at the next ORF.

A different mechanism of translational resistance, relying on the translation of a single uORF in the 5′ leader, has been proposed for *DDIT3* (*Palam et al., 2011*). A fraction of scanning ribosomes recognize and initiate at the uORF initiation codon in a weak Kozak context. Under normal conditions with a high initiation rate, the translation of this uORF inhibits leaky scanning by the obstruction of scanning ribosomes. Under stress conditions, the reduced ribosomal loading results in an alleviation of this obstruction.

For the examples mentioned above, translational control is based on the reduced availability of TC. In specific cases initiation can occur without eIF2. Some viral mRNAs harbour internal ribosome entry sites (IRES) that allow translation initiation to take place by recruiting alternative factors, that is eIF5B (*Pestova et al., 2008*; *Terenin et al., 2008*), eIF2D, and a complex of MCTS1 (a.k.a. MCT-1), and

DENR (*Dmitriev et al., 2010*; *Skabkin et al., 2010*), or even initiate without Met-tRNAi and any initiation factors (*Wilson et al., 2000*). However, the existence of 'viral-like' IRESs in mammalian mRNAs remains controversial (*Shatsky et al., 2010*; *Jackson, 2013*).

The present work uses ribosome profiling (*Ingolia et al., 2009*) to explore the immediate effect of sodium arsenite treatment ($NaAsO_2$), which results in a rapid phosphorylation of eIF2, on protein synthesis. This technique provides a snapshot of translating ribosomes over the entire transcriptome with subcodon resolution (see reviews by *Michel and Baranov, 2013*; *Ingolia, 2014*).

## Results

### Ribosome profiling

In order to generate the most informative conditions for characterizing eIF2-dependent mechanisms of translation regulation, it was important to minimize the transcriptional response and induce significant but not complete inhibition of translation. For this purpose, we chose to treat cells with sodium arsenite for a short time period and to monitor the immediate translational response. Sodium arsenite is a well-known potent inducer of eIF2 phosphorylation that activates EIF2AK1 (*McEwen et al., 2005*). We examined changes in the phosphorylation status of eIF2 and EIF4EBP1 during arsenite treatment to identify suitable conditions for ribosome profiling. The phosphorylated form of eIF2 (p-eIF2) progressively accumulates during the first 2 hr of stress (*Figure 1A*). After 0.5 hr of arsenite treatment, p-eIF2 reaches 30–40% of maximal levels of phosphorylation (*Figure 1A*). The dephosphorylation of EIF4EBP1 becomes evident 1 hr after treatment and does not revert (*Figure 1A*). We did not detect changes in phosphorylation of p70 S6 kinase and its substrate RPS6 during arsenite treatment (*Figure 1A*). A robust accumulation of ATF4 is evident 30 min after treatment. These results suggest that 0.5 hr post-treatment is likely to be suitable for examining eIF2 inhibition whilst minimizing possible arsenite-induced side effects. Furthermore, the number of ribosomes in polysome fractions was reduced by ~4.5-fold under these conditions (*Figure 1B*).

Therefore, HEK293T cells were treated with 40 µM sodium arsenite for 30 min before harvesting for ribosomal profiling which was carried out according to the Ingolia et al. protocol (*Ingolia et al., 2012*) with some modifications (see 'Materials and methods').

*Figure 1C–E* shows the general characteristics of the ribo-seq and mRNA-seq datasets. As expected, in both conditions most ribo-seq reads were mapped to coding regions. The distribution of ribo-seq 5′-end reads, but not of mRNA-seq 5′-end reads (same size randomly fragmented 'naked' mRNA isolated from cytoplasmic lysate), exhibits the characteristic triplet periodicity (*Figure 1—figure supplement 1*). Over 7000 mRNA sequences were uniquely mapped with at least 100 ribo-seq reads.

Owing to the stochastic nature of massively parallel sequencing, the accuracy of an estimate of the level of expression of a gene is dependent on its sequencing depth. Therefore the estimated expression levels of weakly expressed genes have greater variability than highly expressed genes. To mitigate this effect we used a Z-score transformation (see review by *Quackenbush, 2002*). Genes were first ordered based on their lowest read depth (minimum expression). The parameters of the distribution of expression changes for the genes with similar expression levels were used to calculate Z-scores of differential expression for individual genes (see 'Materials and methods' and *Figure 1—figure supplement 2*). We used a Z-score of 4 as an arbitrary threshold of statistical significance for differentially regulated genes to minimize the false discovery rate.

### Effects of arsenite treatment on transcriptome

The estimated time required for mRNA maturation (*Femino et al., 1998*; *Audibert et al., 2002*) is comparable to the duration of the arsenite treatment, therefore we did not expect significant changes in mRNA levels due to a transcriptional response. However, it is conceivable that the arsenite treatment could affect the stability of specific mRNAs. Indeed, the treatment was found to significantly alter the transcript levels of 24 genes (*Figure 1E*). The most pronounced effect observed is the accumulation of JUN mRNA, a transcript that is short-lived under normal conditions (*Elkon et al., 2010*; *Rabani et al., 2011*). JUN encodes a subunit of the AP-1 transcription factor implicated in response to a myriad of physiological and pathological stimuli (reviewed in *Hess et al., 2004*). AP-1 transcription factors consist of homo- or heterodimers of different subunits and its composition is crucial for its specificity. As displayed in *Figure 1F*, despite a significant decrease in its translational

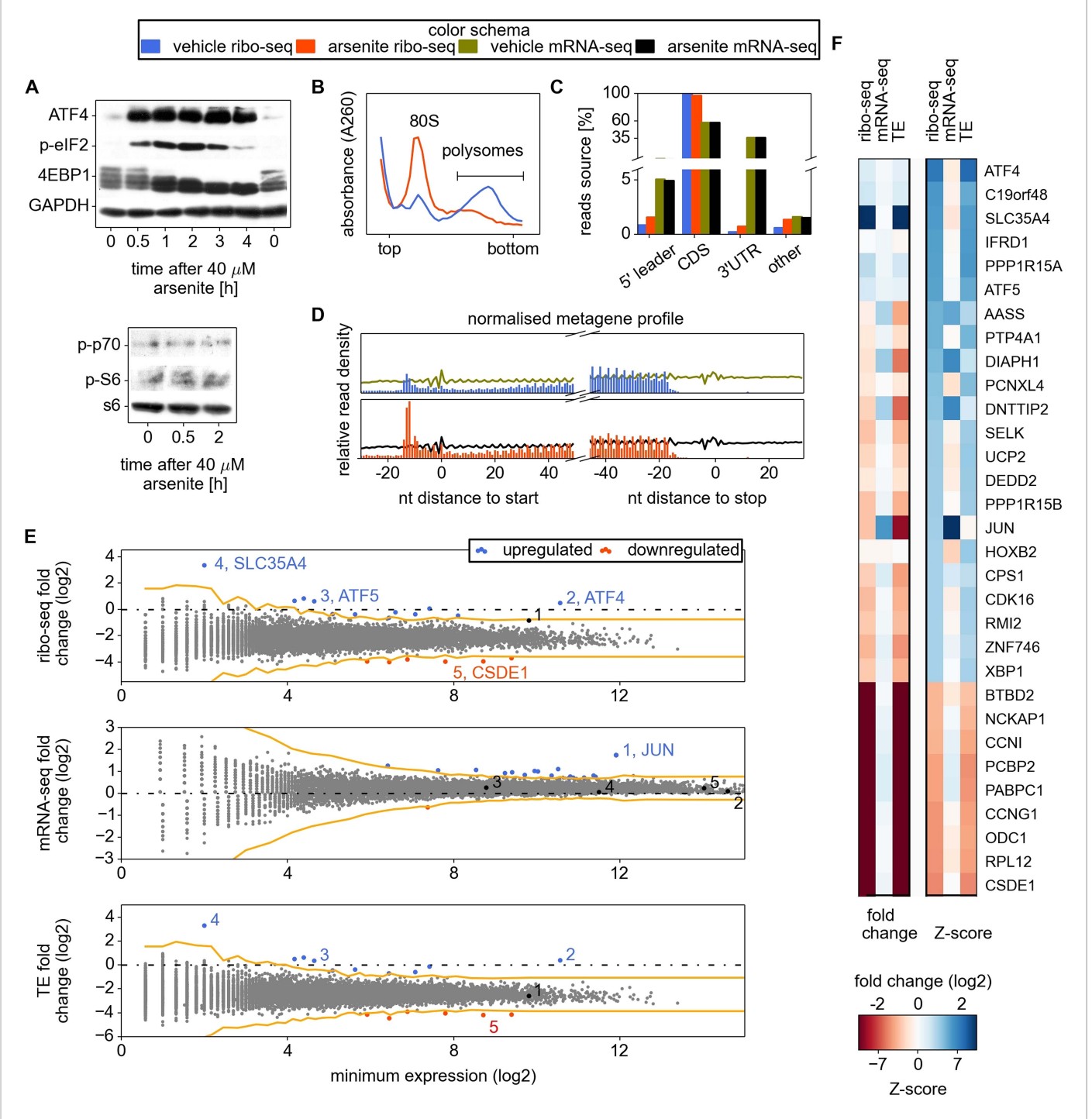

**Figure 1**. Analysis of differential gene expression under conditions of oxidative stress induced with sodium arsenite treatment. (**A**) Western blotting time series analysis of several protein components of HEK293T lysates after treatment with 40 μM sodium arsenite. (**B**) Sucrose density gradient profiles of HEK293T cells untreated and treated for 30 min with sodium arsenite at 40 μM. (**C**) Distribution of raw read counts over mRNA functional regions. (**D**) Metagene analysis: short reads from all mRNAs are aligned around 5′ and 3′ ends of CDS, transcript read density (RNA) is shown using curves, ribosome density (number of footprint reads) is shown using columns corresponding to the alignment locations of the read 5′ ends. (**E**) Differential gene expression analysis. Scatter plots compare ribosome occupancy (top), transcript levels (middle), and translation efficiency (TE) (bottom) between treated and untreated conditions. To avoid error due to uORF translation the number of ribo-seq reads aligning to the CDS only was used to determine the ribosome occupancy and TE. The x axis represents the normalized number of reads corresponding to the experiment/condition of minimal expression (see 'Materials and methods'). The threshold used to denote differentially expressed genes (Z-score of 4) is indicated in orange. Certain genes of interest

*Figure 1. continued on next page*

*Figure 1. Continued*

are indicated with numbers, followed by their gene symbols. (**F**) Two heat maps displaying the fold change and Z-score for the top 22 most stress resistant and bottom 9 most stress sensitive genes, as estimated based on statistical significance of the change of their ribosome occupancy (ribo-seq Z-score).

The following source data and figure supplements are available for figure 1:

**Source data 1**. Read counts and statistics of gene expression response for RNA-seq and ribo-seq experiments for control and stress conditions for each individual transcript.

**Figure supplement 1**. Additional characteristics of ribosome profiling data.

**Figure supplement 2**. Analysis of differential gene expression.

efficiency, the overall expression of JUN is almost unaffected upon arsenite treatment because of the increase in its mRNA levels (*Figure 1E,F*). According to previous observations, AP-1 may induce apoptosis upon treatment with arsenite (*Huang et al., 1999*; *Namgung and Xia, 2000*).

## Arsenite treatment strongly inhibits global translation while translation of a few specific mRNAs is resistant

A median 5.4-fold reduction of ribosomal occupancy (translational efficiency, TE) was observed with the profiling data, a value that is consistent with the reduced number of ribosomes in the polysome fraction (*Figure 1B*). A relatively small subset of mRNAs displayed exceptional sensitivity to translational inhibition (see *Figure 1—source data 1*; the most prominent ones are displayed in *Figure 1F*). Among these is *ODC1* which codes for ornithine decarboxylase, the rate-limiting enzyme of the polyamine biosynthesis pathway. An interesting case of potential downregulation is in *EIF2AK2* (a.k.a. *PKR*), which encodes one of the four eIF2 kinases (we refer to it as potential because its TE and ribo-seq Z-scores did not pass the threshold of statistical significance, but are close to it). Among other extremely sensitive genes are several that encode RNA-binding proteins (*PABPC1*, *PCBP2*, *RPL12*, and *CSDE1*) and cyclins G1 (*CCNG1*) and I (*CCNI*).

To explore the possible activation of the mTOR signalling axis after the course of 0.5 hr arsenite treatment, we analysed the translation of mRNAs that were reported to be strongly downregulated upon pharmacological inhibition of mTOR (*Hsieh et al., 2012*). Almost all of them have negative Z-scores with their TE decrease upon arsenite treatment ~25% greater than the average (*Figure 1—figure supplement 2C*). Thus, while arsenite treatment may affect the mTOR pathway, its impact on translation control is not substantial in comparison with eIF2 inhibition.

Several genes that were previously reported to resist the translation inhibition caused by eIF2 phosphorylation were also found to be resistant in our study (*Figure 1F*). This includes the well-studied *ATF4*, *ATF5*, and *PPP1R15A*. We did not observe translational resistance for either *SRC* (*Figure 1—source data 1*), which was reported to be translated in an eIF2-independent mode (*Allam and Ali, 2010*), or *PRNP*, which was also reported to escape eIF2 associated repression (*Moreno et al., 2012*). The general repression of eIF2 may be expected to promote translation of cellular IRESs if they enabled an eIF2-independent mode of translation as reported for *XIAP* mRNA (*Thakor and Holcik, 2012*). However, we found no evidence of resistance by any genes with putative IRES elements according to the IRESite database (*Mokrejs et al., 2010*), see *Table 1* including XIAP mRNA. It is important to note that many of the genes from the IRESite are not expressed at levels sufficient for detecting resistance.

## Efficient translation of uORFs combined with inefficient translation of CDS is a predictor of stress resistant mRNAs

The mRNAs encoding ATF4, PPP1R15A, SLC35A4, C19orf48, ATF5, and HOXB2 were found to be 'preferentially translated' (defined as having a TE >4 and a fold change >1), while mRNAs encoding IFRD1, PTP4A1, PCNXL4, and UCP2 were found to be 'resistant' (TE >4 and a fold change <1). Due to the small number of preferentially translated and resistant mRNAs, we analysed their properties together and, for simplicity, we refer to them as resistant for the remainder of this text.

**Table 1**. Translation response of mRNAs with reported IRES from IRESite

| Gene_name | IRES name | ORF# | Minimal expression | TE fold change | TE Z-score |
|---|---|---|---|---|---|
| AGTR1 | AT1R_var1 | 1 | 4 | 0.46 | 1.12 |
| AGTR1 | AT1R_var2 | 1 | 4 | 0.46 | 1.12 |
| AGTR1 | AT1R_var3 | 1 | 4 | 0.46 | 1.12 |
| AGTR1 | AT1R_var4 | 1 | 4 | 0.46 | 1.12 |
| APAF1 | Apaf-1 | 1 | 102.9 | 0.29 | 1.8 |
| AQP4 | AQP4 | 1 | 1.5 | 0.98 | 2.47 |
| ATAD5 | ELG1 | 1 | 1083.4 | 0.22 | 0.75 |
| BAG1 | BAG1_p36delta236 nt BAG1_p36 | 4 | 1376.5 | 0.1 | −1.31 |
| BCL2 | BCL2 | 1 | 10 | 0.18 | −0.22 |
| BIRC2 | c-IAP1_285-1399 c-IAP1_1313-1462 | 1 | 147 | 0.13 | −0.94 |
| CCND1 | CCND1 | 1 | 213 | 0.17 | −0.15 |
| CDK11A | PITSLRE_p58 | 1 | 0 | NA | NA |
| CDKN1B | p27kip1 | 1 | 1252 | 0.17 | −0.3 |
| CSDE1 | UNR | 1 | 16,657.5 | 0.05 | −5.12 |
| DCLRE1A | hSNM1 | 1 | 1065.2 | 0.17 | −0.24 |
| EIF4G1 | eIF4G | 1 | 12,937 | 0.21 | 0.65 |
| EIF4G1 | eIF4GI-ext | 1 | 12,937 | 0.21 | 0.65 |
| EIF4G2 | DAP5 | 1 | 21,727.6 | 0.26 | 1.56 |
| EIF4G3 | eiF4GII | 1 | 1305.6 | 0.13 | −1.46 |
| EIF4G3 | eIF4GII-long | 1 | 1305.6 | 0.13 | −1.46 |
| FGF1 | FGF1A | 1 | 0 | NA | NA |
| FMR1 | FMR1 | 1 | 137.8 | 0.13 | −1.35 |
| HSPA1A | hsp70 | 1 | 0 | NA | NA |
| HSPA5 | BiP_-222_-3 | 1 | 1819.3 | 0.23 | 1 |
| IGF2 | IGF2_leader2 | 1 | 0 | NA | NA |
| LAMB1 | LamB1_-335_-1 | 1 | 1141 | 0.13 | −1.52 |
| LEF1 | LEF1 | 1 | 248 | 0.15 | −0.8 |
| MNT | MNT_75-267 MNT_36-160 | 1 | 144 | 0.28 | 1.32 |
| MYB | MYB | 1 | 154.2 | 0.09 | −1.47 |
| MYC | c-myc | 2 | 1946 | 0.14 | −1.2 |
| MYCL1 | L-myc | 1 | 0.5 | NA | NA |
| MYCN | n-MYC | 1 | 0 | NA | NA |
| NKRF | NRF_-653_-17 | 1 | 1064 | 0.18 | −0.01 |
| PDGFB | PDGF2/c-sis | 1 | 0 | NA | NA |
| PIM1 | Pim-1 | 1 | 113 | 0.12 | −1.1 |
| RUNX1 | AML1/RUNX1 | 1 | 10 | 0.16 | −0.38 |
| RUNX1T1 | MTG8a | 1 | 163 | 0.13 | −1.08 |
| XIAP | xIAP_5-464 XIAP_305-466 | 1 | 2169.6 | 0.12 | −1.67 |

Examination of individual mRNA profiles frequently revealed the presence of extensively translated uORFs in resistant mRNAs (*Figure 2*). Indeed, with the exception of a single weakly expressed gene *HOXB2*, all mRNAs found to be resistant (TE Z-score >4) contained uORF(s) that are translated under normal conditions. However, ribosome profiling data suggest that 8% of other expressed mRNAs also

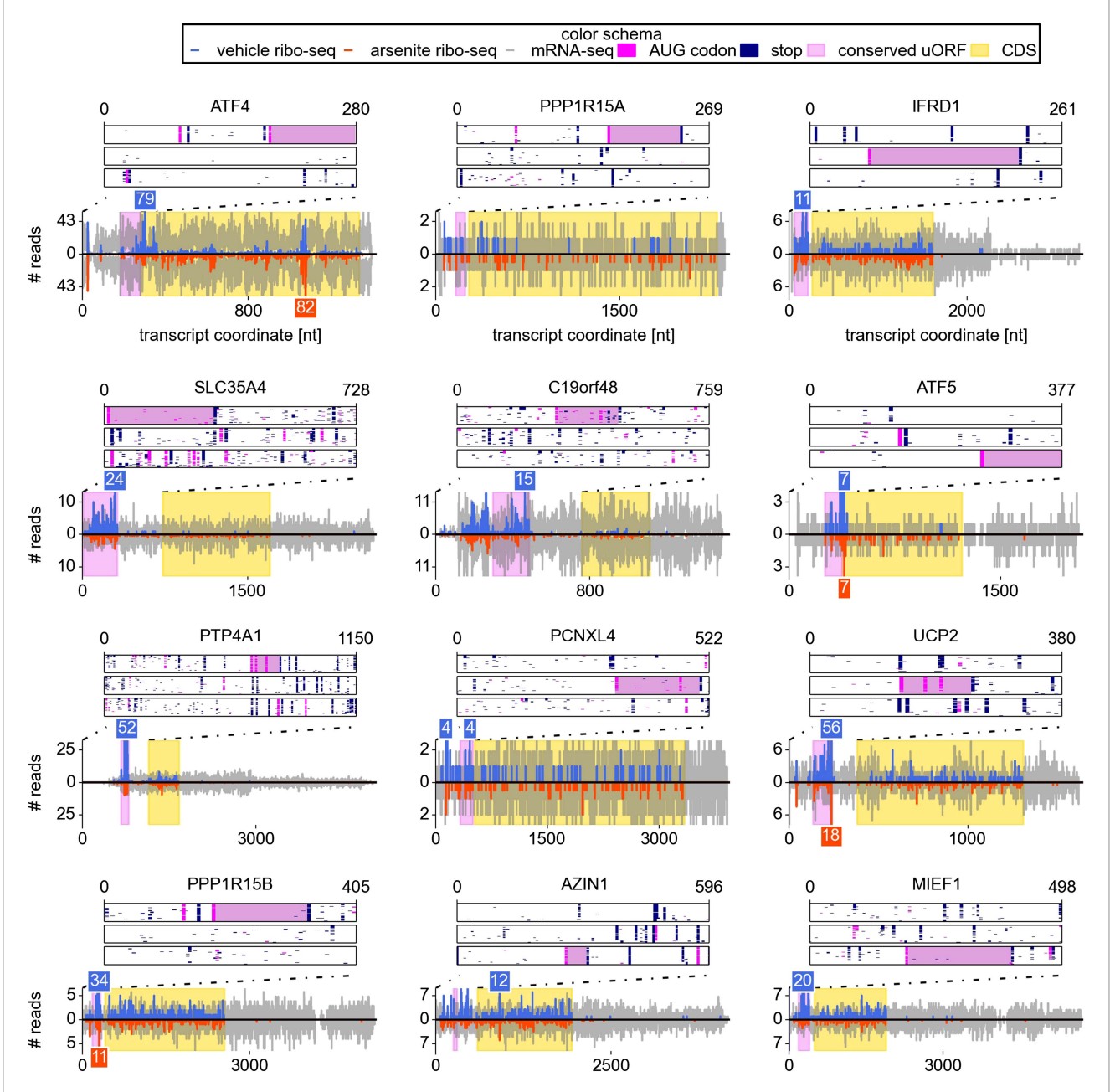

**Figure 2**. Upstream open reading frame (uORF) conservation and ribosome density for the eight top most stress resistant mRNAs in terms of their translation efficiency and also for mRNAs of UCP2, PPP1R15B, AZIN1, and MIEF1. Bottom plots for each mRNA show counts of mRNA-seq reads (grey) and ribosome reads (blue and red) as columns (control: positive values; arsenite treatment: negative values). The annotated CDS region is highlighted in yellow. Translated conserved ORFs in the 5′ leaders are highlighted in violet. Read counts above the cut-off are shown with numbers above corresponding columns. Top plots represent conservation of uORF features within the leaders of the orthologous mRNAs (upstream of annotated CDS) obtained from the analysis of genomic alignments of the 46 vertebrates using the human sequence as a reference. Each box corresponds to one of the three reading frames where AUG codons are shown as pink dots and stop codons as navy dots in each of the genomic sequences used in the alignments. Regions of multiple sequence alignment corresponding to translated conserved uORFs are highlighted in violet. Introns and gaps were removed from the alignments.

The following figure supplements are available for figure 2:

**Figure supplement 1**. Multiple alignments of codon sequences from 100 vertebrate genomes aligned to the region of the conserved *SLC35A4* upstream open reading frame (uORF) in three different frames.

*Figure 2. continued on next page*

*Figure 2. Continued*

**Figure supplement 2**. Multiple alignments of codon sequences from 100 vertebrate genomes aligned to the region of conserved MIEF1 upstream open reading frame (uORF) in three different frames.

**Figure supplement 3**. Publicly available ribosome profiling data in GWIPS-viz for *SLC35A4* and *MIEF1*.

have translated uORFs (*Figure 3A*). Hence, the mere presence of translated uORFs is a poor predictor of resistance. We therefore investigated further the features of uORFs that are associated with stress resistance. As can be seen in *Figure 3B*, uORFs in the resistant mRNAs are usually efficiently translated under normal conditions, though yet again there is a large absolute number of non-resistant mRNAs that contain similarly efficiently translated uORFs. *Figure 3C* shows the relationship between the TE of the CDS and the resistance: the CDS of the most resistant mRNAs is weakly translated under normal conditions. The ratio of the ribosome densities in uORFs and in CDS provides a much better criterion for discriminating between resistant and non-resistant mRNAs (*Figure 3D*). The length of all the translated uORFs (in the resistant mRNAs) was found to exceed 20 codons, although this is of limited predictive value as there are many long uORFs in non-resistant

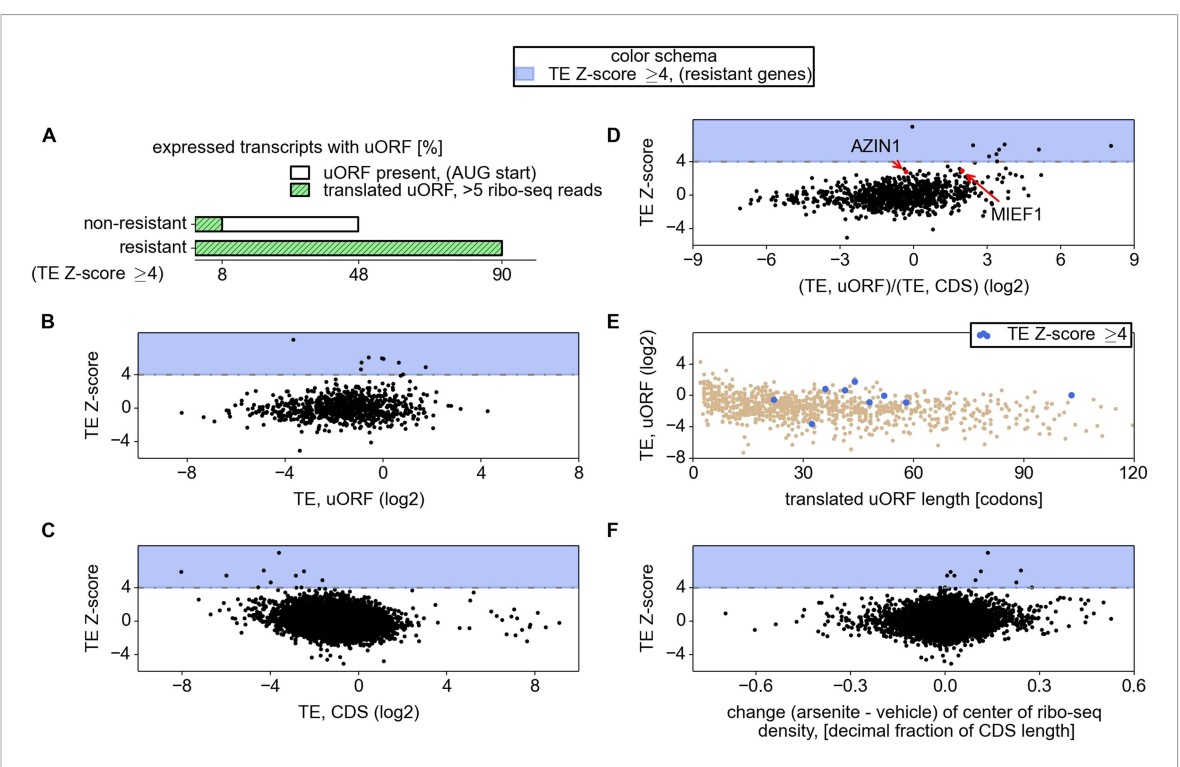

**Figure 3**. Relationship between mRNA stress resistance and upstream open reading frames (uORFs). (**A**) Frequency of AUG initiating uORF occurrence and their translation in stress resistant and other mRNAs. Relationship between stress resistance (*y* axis) and translation efficiency of uORFs (**B**), CDS (**C**), and uORF/CDS ratio (**D**). Translationally resistant genes (shaded in blue) have a high uORF translational efficiency (TE) and a low CDS TE. (**E**) Relationship between uORF length (*x* axis), their TE (*y* axis), and the level of stress resistance (differential colouring). (**F**) Relationship between stress resistance (*y* axis) and shift of ribosome density in the 3′ direction. These plots indicate that, under normal conditions, resistant mRNAs tend to display a high uORF TE and a low CDS TE while, under stress conditions, resistant mRNAs are associated with a shift of ribosome density in the 3′ direction owing to a reduced ratio of ribosome density between uORFs and CDS.

The following figure supplement is available for figure 3:

**Figure supplement 1**. Analysis of 5′ leader and upstream open reading frame (uORF) features in the resistant mRNAs.

mRNAs (*Figure 3E*). Based on these findings we expected that, upon arsenite treatment, the ribosome density for resistant mRNAs would shift from the 5′ leaders to CDS. Such a shift is indeed observable (*Figure 3F*).

We also compared various sequence features of 5′ leaders and uORFs between the resistant mRNAs and the remaining expressed mRNAs. We explored the nucleotide (nt) context surrounding uORF start codons (mostly AUG but also CUG) in resistant mRNAs but found no evidence for selection for a particular context (*Figure 3—figure supplement 1A*); the frequency of individual heptameric initiation sequences (−3 to +4) is equally variable across uORFs of resistant and non-resistant mRNAs as well as at annotated starts of CDS (*Figure 3—figure supplement 1B*). We found that the average 5′ leader length of resistant mRNAs is longer (378.5 nt) than that of other mRNAs (169.0 nt), but there is significant variation within both distributions (*Figure 3—figure supplement 1C*). We also found that 5′ leaders of resistant mRNAs have lower potential for RNA secondary structure formation within the first 240 nt based on free energy estimates of potential structures predicted with RNAfold (*Lorenz et al., 2011*). Yet, the difference is small and RNA secondary structure potential does not correlate well with resistance (*Figure 3—figure supplement 1D*).

Based on this analysis we concluded that uORFs are ubiquitous in all highly resistant mRNAs expressed in HEK293T cells and the efficient (and perhaps inhibitory) translation of these uORFs most likely plays a crucial role in the mechanism of resistance. The mere presence of an uORF and its translation is insufficient to provide the resistance to eIF2 inactivation.

## Newly discovered cases of resistance to eIF2 inhibition mediated by uORFs

We focused our attention on newly identified uORF-bearing mRNAs whose translation was refractory to eIF2 inactivation. For some of these the regulatory function of uORFs has been described previously, but its implication in eIF2-dependent translational control was not shown. This was true for *IFRD1* (a.k.a. *PC4* and *TIS7*), an interferon-related developmental regulator that was reported to be a modifier gene for cystic fibrosis (*Gu et al., 2009*). It has been reported previously that one of the two *IFRD1* transcript variants possesses a 51 codon uORF which triggers mRNA decay upon termination under normal conditions but not under conditions of Unfolded Protein Response mediated by tunicamycin (*Zhao et al., 2010*). We did not observe a significant change in the transcript level upon arsenite treatment. A second case is *UCP2*, which codes for a mitochondrial anion carrier protein that increases the proton conductance of the mitochondrial membrane in response to reactive oxygen species production (*Baird and Wek, 2012*). It was shown that expression of *UCP2* is upregulated at the translational level upon oxidative stress (*Adam et al., 2006*). The *UCP2* 5′ leader contains a 36 codon uORF that inhibits translation of *UCP2* mRNA under normal conditions (*Hurtaud et al., 2006*).

To our knowledge the other newly identified mRNAs have not been shown to be regulated at the translational level before. One of the most unusual examples is the mRNA encoding the probable UDP-sugar transporter protein SLC35A4 (*Figure 2*). Its 5′ leader contains 11 AUG codons, most of which are not conserved; however, one AUG that initiates a 102 codon uORF is highly conserved across vertebrates. This uORF encodes a peptide sequence containing PFAM domain DUF4535, ID PF15054; moreover, the pattern of its conservation is consistent with protein coding evolution (*Figure 2—figure supplement 1*), suggesting that this uORF likely encodes a functional protein. This alternative protein (EMBL accession HF548106) was recently detected by mass spectrometry analysis of cultured cells and human tissues (*Vanderperre et al., 2013*; *Kim et al., 2014*). We examined translation of this mRNA in other publicly available ribosome profiling datasets using GWIPS-viz (*Michel et al., 2014*) and found that this uORF is translated in all datasets (*Figure 2—figure supplement 3A*). How ribosomes reach the 12th AUG codon upon arsenite treatment is unclear and merits further investigation.

Notably, one of the resistant mRNAs found in our study is the *PPP1R15B* gene that encodes a phosphatase that dephosphorylates eIF2, PPP1R15B (a.k.a. CReP) (*Novoa et al., 2001*). Sustained translation of PPP1R15B mRNA under conditions of eIF2 inactivation represents a feedback loop for reactivation of eIF2 during recovery from stress.

Although they did not pass our stringent criteria for a resistant gene, other candidates which we identified (based on the gene function and their profiles) are *AZIN1* (TE Z-score 2.76) and *MIEF1*

(TE Z-score 2.88). *AZIN1* encodes an inhibitor of ODC1 (ornithine decarboxylase) antizymes. Antizymes are proteins that target ODC1 for degradation, and AZIN1 is highly similar to ODC1 but lacks ornithine decarboxylation enzymatic activity. This makes it a competitive inhibitor of antizymes (*Murakami et al., 1988*). It has been shown that an uORF initiated with a non-cognate AUU codon mediates sensitivity of *AZIN1* mRNA translation to polyamine levels (*Ivanov et al., 2008*). *MIEF1*, mitochondrial elongation factor 1, is another candidate bicistronic mRNA that we have identified (the other is *SLC35A4*). Similar to *SLC35A4*, we observed evidence of protein coding evolution within its uORF (*Figure 2—figure supplement 2*). Its translation is also supported by multiple ribo-seq datasets available in GWIPS-viz (*Michel et al., 2014*), see *Figure 2—figure supplement 3B*. Examination of the sequence encoded by its uORF revealed that it contains a conserved domain that belongs to a PFAM family Complex 1 protein (LYR family), ID PF05347.

## 5′ leaders of several newly identified mRNAs are sufficient to provide resistance to translation inhibition

To examine the role of the 5′ leaders in modulating the resistance to eIF2 inhibition of resistant mRNAs revealed in this study (*IFRD1*, *PPP1R15B*, *UCP2*, *PTP4A1*, and *SLC35A4*), we designed reporter constructs and prepared capped and polyadenylated mRNAs with sequences of 5′ leaders upstream of a Firefly luciferase (Fluc) coding region. We used the 5′ leader of *ATF4* mRNA and the HCV IRES as positive controls and, as a negative control, we used a non-specific 63 nt leader from the vector pGL3. mRNAs along with a control mRNA encoding *Renilla* luciferase (Rluc) were transfected into HEK293T cells and simultaneously treated with 40 μM arsenite or vehicle (*Andreev et al., 2009*).

Under normal conditions the translation of mRNAs bearing the 5′ leaders of *IFRD1*, *PPP1R15B*, *UCP2*, and *PTP4A1* was about sevenfold lower than that of the control mRNA with the simple non-specific leader (pGL3), whereas *SLC35A4* was even lower (*Figure 4A* and *Figure 4—figure supplement 1A*). Arsenite treatment resulted in significant inhibition of pGL3 and control Rluc translation, while translation of other mRNAs did not change considerably and even slightly increased for SLC35A4 and HCV IRES. Similar results were observed in the Huh7 hepatocarcinoma cell line (*Figure 4—figure supplement 1C*). To address the effect of arsenite treatment on ongoing translation, which may be more relevant than conditions applied for ribosome profiling, reporter mRNAs were transfected and 1 hr later cells were treated with either non-specific translational inhibitor cycloheximide or arsenite. As expected, both inhibitors efficiently blocked translation of control Rluc mRNA but only cycloheximide was able to arrest translation driven by leaders of resistant mRNAs. Surprisingly, during arsenite treatment the reporter mRNA with the SLC35A4 5′ leader was able to produce 15 times more luciferase than after treatment with cycloheximide (*Figure 4—figure supplement 2*).

For several other reporter mRNAs with different 5′ leaders which possess low TEs under normal conditions, translation was significantly downregulated upon arsenite treatment (data not shown). This rules out the possibility that low TE of reporter mRNA results in resistance to stress.

We also measured the kinetics of protein synthesis to rule out the possibility that our observations can be explained by mRNA silencing or destabilization (*Andreev et al., 2013*). For this purpose cells were treated with arsenite (or a vehicle) immediately after transfection and luciferase activity was measured over time. Both *IFRD1* and *PPP1R15B* reporters showed that luciferase activity increases over time with no indication of a plateau that would be expected upon mRNA destabilization (*Figure 4B*).

Treatment with a potent inhibitor of mTOR kinase, torin-1 (*Thoreen et al., 2009*), led to inhibited translation of these reporters to the same degree as the control pGL3 mRNA (*Figure 4—figure supplement 3*). Thus, *IFRD1* or *PPP1R15B* leaders do not provide translational resistance to the stress response that involves sequestration of the cap-binding protein eIF4E.

It was conceivable that translational resistance was due to side effects of arsenite treatment rather than its direct effect of eIF2 inactivation. To directly address the impact of eIF2 phosphorylation on reporter mRNA translation during arsenite stress, we carried out an experiment where cells were pre-transfected with a plasmid encoding the full length human PPP1R15A (a.k.a. GADD34) phosphatase subunit which is able to reverse eIF2 phosphorylation (*Brush et al., 2003*). Arsenite-induced eIF2 phosphorylation, as expected, was almost completely alleviated in the presence of GADD34 (residual phosphorylation probably reflects less than 100% efficient plasmid transfection), see *Figure 4—figure supplement 4*. As a result, the downregulation of control Rluc mRNA was only twofold in comparison

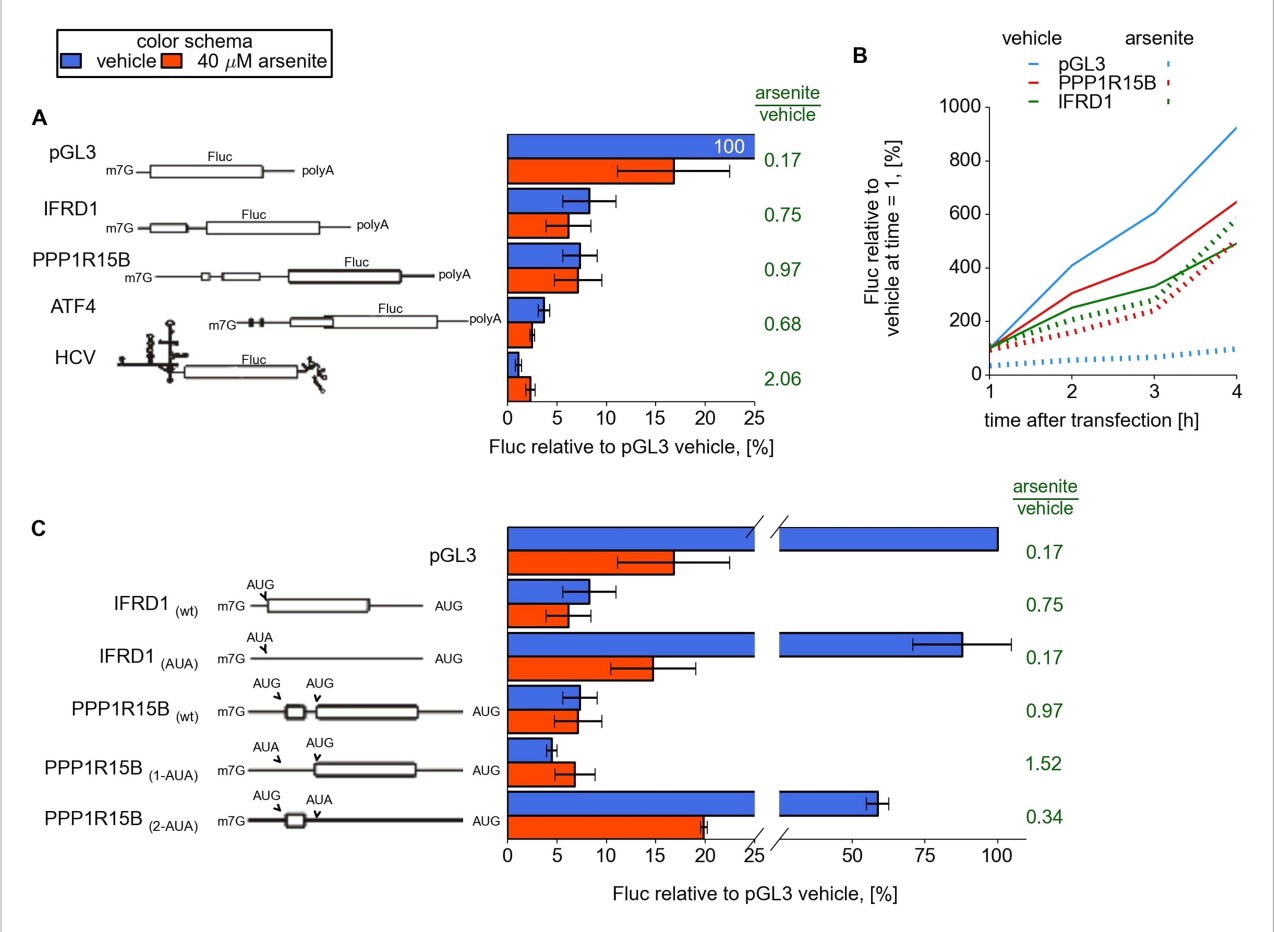

**Figure 4**. Upstream open reading frame (uORF) involvement in modulation of IFRD1 and PPP1R15B mRNAs stress resistance. (**A**) Firefly luciferase (Fluc) activity produced by expression of mRNA containing different 5′ leaders 2 hr after arsenite treatment (red bars) and in untreated cells (blue bars). Relative units correspond to Fluc activity normalized to the median Renilla luciferase (Rluc) activity derived from a co-transfected Rluc mRNA. The green text represents fold change calculated from the same data. The ORF organization of examined mRNAs is outlined on the left. Bars represent standard deviations. (**B**) Time series analysis of Fluc expression in cells treated at the time of transfection with sodium arsenite to a concentration of 40 μM (dotted lines) or vehicle (solid lines). Fluc activity of vehicle at 1 hr was taken as 100% for each mRNA and experimental condition. (**C**) Effect of start codon identity in IFRD1 and PPP1R15B 5′ leaders on Fluc activity.

The following figure supplements are available for figure 4:

**Figure supplement 1**. (**A**) Resistance of additional reporters with 5′ leaders of ATF5, UCP2, PTP4A1, and SLC35A4 to arsenite treatment.

**Figure supplement 2**. Effect of arsenite treatment on ongoing reporter translation.

**Figure supplement 3**. (**A**) Effect mutations that improve initiation Kozak context for uAUG in the IFRD1 leader during arsenite treatment.

**Figure supplement 4**. Top: Effect of GADD34-Flag (PPP1R15A) overexpression on activity of firefly luciferase under control of the IFRD1 mRNA leader (Fluc, green bars) and on mRNA encoding *Renilla* luciferase (Rluc, light pink bars).

with a more than sixfold reduction in cells not transfected with the GADD34 plasmid. Translation of the IFRD1 reporter was not affected under either condition. We therefore concluded that translational inhibition caused by arsenite treatment is predominantly due to the phosphorylation of eIF2. Also, we treated cells with 2.5 mM dithiothreitol (DTT), triggers the Unfolded Protein Response and results in eIF2 phosphorylation (*Prostko et al., 1993*). We found that the leaders of IFRD1 and PPP1R15B provide translational resistance under these conditions as well (*Figure 4—figure supplement 1B*).

## Site-specific mutagenesis confirms the critical role of *IFRD1* and *PPP1R15B* uORF translation in mediating resistance to eIF2 inhibition

While both *IFRD1* and *PPP1R15B* mRNAs possess uORFs with high TE (*Figure 2*), the architecture of their uORFs is markedly different (schematic organization of the 5′ leaders is depicted in *Figure 4*). *IFRD1* mRNA contains a single highly conserved 53 codon uORF that starts 19 nt from the 5′ end and ends 43 nt upstream of the main ORF. *PPP1R15B* mRNA contains two in-frame uORFs separated by 21 nt. The first uORF is eight codons long and is 127 nt from the 5′ end. The second 52 codon long uORF is 75 nt upstream of the CDS.

Substitution of the *IFRD1* uORF AUG with the AUA codon increased the reporter expression eightfold but made the translation susceptible to eIF2 inhibition. For *PPP1R15B*, substitution of the first uORF AUG with AUA slightly reduced the reporter activity under normal conditions but, surprisingly, further increased the reporter resistance to the stress. A similar substitution of the second uORF start codon significantly reduced the resistance to stress, as in the case with *IFRD1* (*Figure 4C*).

We conclude that neither of these genes is regulated by delayed reinitiation (as in *ATF4* and *GCN4*), since in both cases a single uORF is sufficient for eIF2-mediated translational control.

A single uAUG in *IFRD1* mRNA is in a suboptimal initiation context (A in −3 position but U in + 4). To explore how the context may affect the resistance we introduced +4U/G mutation that improves the context. We found a slight inhibition of the *IFDR1* main ORF translation under normal conditions (presumably due to an increased inhibitory effect of the uORF translation). However, this mutation did not alter the sensitivity of the main ORF translation to arsenite stress (*Figure 4—figure supplement 3A*).

## An unstructured leader sequence upstream of IFRD1 uORF is necessary for stress resistance

We observed earlier that most stress resistant mRNAs possess efficiently translated uORFs. We hypothesized that some features of the 5′ leaders upstream of uORFs may be important for resistance. To address this issue we created two additional reporters based on control pGL3 and IFRD1, where we added a 5′ terminal stem loop of intermediate stability (*Figure 5*). As expected, the addition of this stem loop resulted in a threefold to fourfold decrease in the activity of both reporters under normal conditions. Interestingly, when arsenite stress was induced, the SL-IFRD1 construct did not exhibit resistance while translation of SL-PGL3 was downregulated as much as the pGL3 construct. Therefore we propose that efficient loading of pre-initiator complexes to the uORF is necessary for stress resistance in IFRD1. Next we addressed the question of whether the specific sequence upstream of the IFRD1 uORF is required for regulation. We substituted it with an artificial single stranded $(CAA)_6$ sequence of the same length. This mutation did not alter stress resistance. Thus, we hypothesize that, for resistant translation, the uORF has to be preceded with a leader allowing a high initiation rate at the uORF.

## Discussion

eIF2 phosphorylation is a key event in the response of cells to various stresses and is also involved in the cell cycle. The global downregulation of protein synthesis triggered by eIF2 inactivation has two main purposes. The first is to conserve cellular resources, and the second is to provide a delay to evaluate the severity of the damage and, depending on its level, reprogram gene expression either towards apoptosis or to a pro-survival repair response. This necessarily requires activation of genes involved in the ISR. mRNAs of genes involved in the ISR ought to be translated under conditions of eIF2 inactivation (see reviews by *Luo et al., 1995*; *Baird and Wek, 2012*).

In order to identify such mRNAs we utilized ribosome profiling, a technique that generates a snapshot of ribosome locations on the entire set of mRNAs (*Ingolia et al., 2009*). We applied this technique to HEK293T cells 30 min after treatment with sodium arsenite, a well-known inducer of eIF2 phosphorylation. This enabled us to study the early stress response at the level of translation. Under our strict criteria of statistical significance (Z-score TE >4), translation of 10 mRNAs was found to be resistant. Translation of six mRNAs (encoding ATF4, PPP1R15A, SLC35A4, C19orf48, ATF5, HOXB2) was increased and translation of four (encoding IFRD1, PTP4A1, PCNXL4, UCP2) was reduced only slightly in comparison with a global reduction in translation. Seven mRNAs (encoding CCNG1, CCNI,

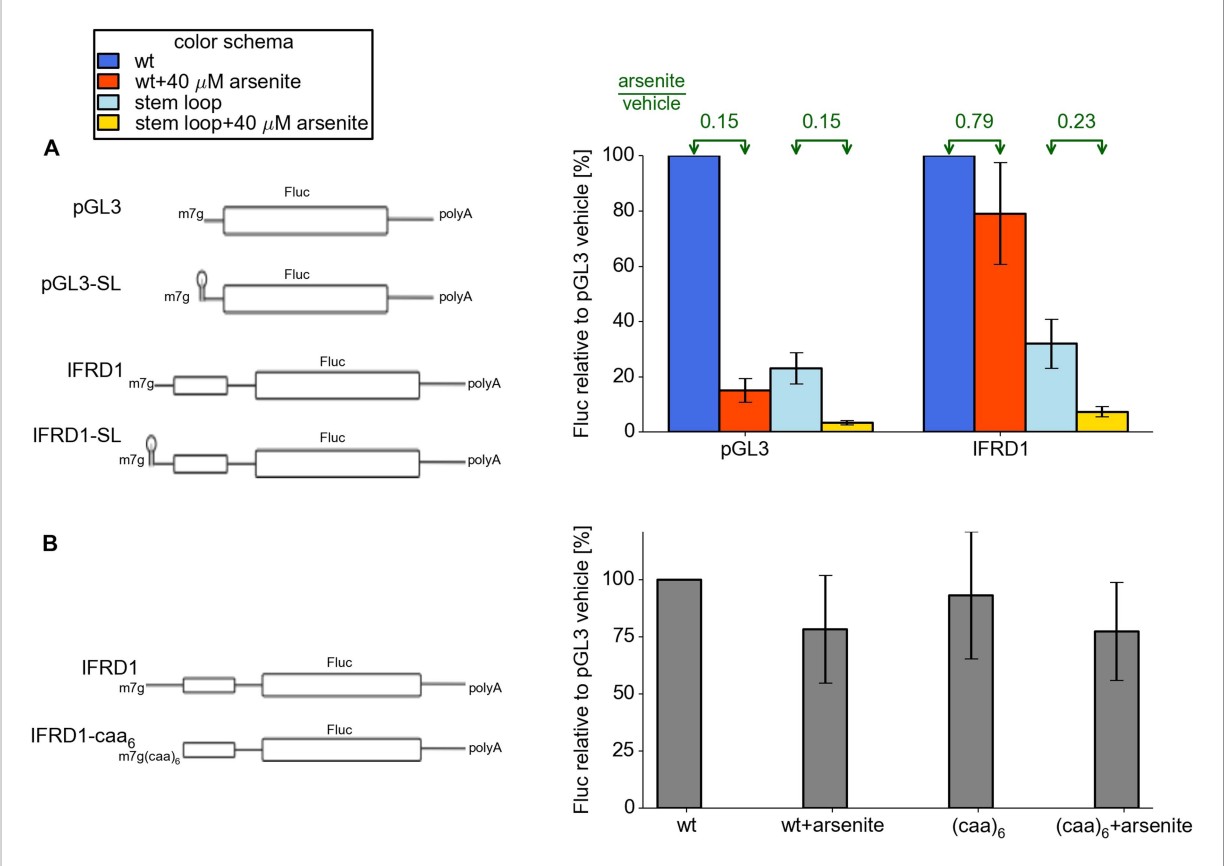

**Figure 5**. Features of the IFRD1 5′ leader required for resistance. (**A**) Firefly luciferase (Fluc) activity produced by expression of mRNA containing pGL3 and IRFD1 leaders (outlined on the left) with and without an additional stem loop at the 5′ end under different conditions. Blue (normal conditions) and red (stress conditions) bars correspond to leaders lacking the stem loop while light blue (normal) and yellow (stress) correspond to leaders with the stem loop. The fold change of Fluc activity in response to stress is indicated above by green arrows. (**B**) Effect of $(CAA)_6$ addition to the 5′ leader of IFRD1 on Fluc activity in response to stress.

CSDE1, ODC1, PABPC1, PCBP2, RPL12) were found to be particularly sensitive. We confirmed translational resistance of some genes previously reported and identified novel stress resistant genes.

Features found to be common amongst the resistant mRNAs are their low levels of CDS translation (*Figure 3C*) and low levels of transcription, except ATF4 (*Figure 1E*). It is unclear, however, whether these features are common due to the resistance-providing mechanism or due to their function. mRNAs resistant to eIF2 are expected to encode components of cell signalling (e.g. transcription factors, kinases, phosphatases) and therefore are not required in large quantities. We also found that, with one exception, all mRNAs significantly resistant to the stress conditions possess translated uORFs in their 5′ leaders. Their number, mutual organization, and depth of phylogenetic conservation vary. *IFRD1* has only a single uAUG in its 5′ leader while *SLC35A4* has 11 (*Figure 2*). For these mRNAs at least one of the uORFs is efficiently translated under normal conditions and is longer than 20 codons. We confirmed that 5′ leaders of some of these mRNAs confer translation resistance to a reporter in synthetic RNA constructs. Owing to the diversity of the uORF organisation in these 5′ leaders, the mechanism of uORF-mediated resistance may vary and should be studied individually. We chose to explore how the sequence properties of *IFRD1* and *PPP1R15B* 5′ leaders affect the resistance of downstream ORFs. For this purpose we carried out site-directed mutagenesis of 5′ leader sequences in the constructs containing a luciferase reporter. We found that translation of only one uORF is sufficient to provide resistance to eIF2 inhibition for *IFRD1* and *PPP1R15B*. Therefore it is likely that the resistance for these transcripts is provided by the mechanism that resembles alleviation of scanning ribosomes obstruction rather than delayed reinitiation (ATF4-like cases). The substitution of IFRD1 leader upstream of an uORF with an artificial single stranded sequence does not affect stress

resistance, ruling out the requirement for a specific nucleotide sequence. However it is likely to enable rapid ribosome loading at the uORF since the addition of a 5′ terminal stem loop of intermediate stability abolished stress resistance. This observation may explain why translation of many mammalian mRNAs possessing uORFs is not resistant to eIF2 phosphorylation. Translation of these uORFs might be inhibited to such an extent that they would be unable to provide any resistance.

While it is possible that uORFs provide mRNAs with stress resistance in the same manner in all resistant mRNAs detected here, we think it is more likely that uORFs are an essential component of the diverse mechanisms. We mentioned earlier two such mechanisms, delayed initiation and alleviation of scanning ribosomes obstruction. However, even an IRES-mediated resistance would likely require translation of an uORF as it would prevent IRES structure melting by the scanning ribosomes.

The genes that we found to be resistant to eIF2 inhibition may participate in the ISR. This is the case with *PPP1R15B*. Similar to *PPP1R15A*, it encodes a subunit of the phosphatase that dephosphorylates eIF2, thus providing feedback preventing complete translation suppression and also enabling recovery from the stress-induced translational arrest (*He et al., 1997*; *Jousse et al., 2003*). Expression of *PPP1R15A* is tightly regulated, its basal level of expression is almost undetectable (*Novoa et al., 2001*), whereas *PPP1R15B* is constitutively present in cells (*Jousse et al., 2003*). PPP1R15B is a short-lived protein whose half-life is approximately 45 min (*Jousse et al., 2003*). Therefore it needs to be continuously synthesized in order to dephosphorylate eIF2. It has previously been found to remain present in the cell upon arsenite and tunicamycin treatments (*Jousse et al., 2003*).

It is also possible that some of the translationally resistant genes are not directly implicated in the ISR. Their resistance could be related to other eIF2-mediated regulatory mechanisms, for example during cell-cycle progression or development (*Datta et al., 1999*; *Harding et al., 2009*). At least one of the newly identified stress resistant mRNAs encoding oncogenic phosphatase PTP4A1 (a.k.a. PRL-1) may be directly implicated in malignant transformation since it downregulates expression of p53 tumour suppressor (*Min et al., 2009*).

Translational resistance of some genes to eIF2 phosphorylation may also be a consequence of the ORF organisation of their mRNAs which serves a different purpose. This could be the case for the candidate bicistronic mRNAs identified in this study (*MIEF1* and *SLC35A4*). The ratio between uORF and main ORF translation changes in both upon eIF2 inactivation. Coding for two functionally related proteins in the same mRNA may be advantageous for coordination of their expression.

To summarize, our work expands the list of mRNAs which are known to be persistently translated under conditions of eIF2 phosphorylation, although it suggests that the number of such mRNAs is very low. The analysis of ribosome densities on mRNAs resistant to eIF2 phosphorylation accentuates the vital role of uORF translation in providing the resistance.

## Materials and methods

### Ribosome profiling

The ribosomal profiling technique was carried out according to *Ingolia et al. (2012)* but with important modifications described below. HEK293T cells were grown in DMEM supplemented with alanyl-glutamine and 10% FBS and replated to 150 mm dishes (two dishes per sample). After the cells reached 70–80% confluency, sodium arsenite (or vehicle) was added at 40 µM and 30 min later the cells were harvested: dishes were immediately chilled on ice and washed with PBS + cycloheximide (100 µg/ml). Importantly, cells were not pre-treated with cycloheximide to avoid artificial accumulation of initiation complexes at translation initiation starts (*Gerashchenko and Gladyshev, 2015*). Cells were then lysed with buffer containing 20 mM Tris–HCl (pH 7.5), 250 mM NaCl, 1.5 mM MgCl$_2$, 1 mM DTT, 0.5% Triton X-100, 100 µg/ml cycloheximide (Sigma-Aldrich), 20 U/ml TURBO DNAse (Ambion, Waltham, MA). Note that the buffer contains low magnesium (1.5 mM), since high magnesium concentrations stabilize secondary structures in mRNAs which may hamper digestion by RNAse I (RNAse I itself does not require divalent cations for its activity) and lowering magnesium concentration in 5 mM to 15 mM range has been shown to improve footprint resolution (*Ingolia et al., 2012*). Cell lysates were incubated on ice for 10 min, centrifuged at 16,000×*g* at +4°C for 10 min, and the supernatant was divided into two parts for ribo-seq and 'naked' mRNA-seq library preparation. One part of the lysate, usually 10–20 A260, was treated with RNAse I (Ambion) with 100 U per 3.1 A260 of lysate at 23°C for 50 min. The digestion was then stopped with the appropriate amount of SUPERASE inhibitor (Ambion). The treated lysate was loaded on 10–60% (m/v) sucrose density gradient containing 20 mM Tris–HCl (pH 7.5), 250 mM NaCl,

15 mM MgCl$_2$, 1 mM DTT, 100 µg/ml cycloheximide and centrifuged in SW-41 rotor at 35,000 rpm for 3 hr. Sucrose density gradients were prepared as described previously (*Stone, 1974*). Briefly, 5.5 ml of 10% sucrose was slowly layered onto the same volume of 60% sucrose, gradient tubes were then sealed with parafilm, slowly placed horizontally for 4 hr to allow spontaneous gradient formation and then slowly returned to a vertical position. This method of gradient formation is highly reliable and reproducible and does not require any special equipment. Total RNA from the fractions corresponding to 80S peak was then extracted with phenol/chloroform followed by ethanol precipitation.

For the mRNA-seq control, the second lysate aliquot was processed with Trizol-LS (Life Technologies, Waltham, MA) according to the manufacturer's protocol. mRNA from total RNA was isolated using the Oligotex mRNA kit (Qiagen, Netherlands). Two rounds of polyA(+)-mRNA selection instead of one were applied to decrease rRNA contamination to approximately 3%. Purified mRNA was then subjected to alkaline hydrolysis as described by *Ingolia et al. (2009)*.

Both ribo-seq and mRNA-seq samples were loaded onto a 15% denaturing urea PAGE (containing 1× TBE, 7 M urea, and AA:bis-AA in the ratio 20:1). Bands corresponding to nucleic acid fragments of 28–34 nt were excised for both ribo-seq and mRNA-seq samples. RNA was extracted by overnight incubation in a shaker using buffer containing 0.3 M NaOAc (pH 5.1), 1 mM EDTA, and 0.1% SDS followed by precipitation with one volume of isopropanol and 2 µl of GlycoBlue (Life Technologies).

The same quantity of both ribo-seq and mRNA-seq fragments (usually 100 ng) were mixed with 1:10,000 of unspecific RNA oligonucleotide 5′AUGUACACGGAGUCGACCCGCAACGCGA 3′ which serves as a 'spike-in' control. The library preparation was carried out as previously described (*Ingolia et al., 2012*) with the following modifications. First, the circularization reaction was performed for 2 hr. Second, during PCR library amplification, the temperature ramping speed was set as 2.2°C/s to reduce bias associated with GC content (*Aird et al., 2011*).

Two independent biological replicates were carried out. Libraries were sequenced either on an Illumina MiSeq genome analyser at the TrinSeq genomic facility (Dublin) or on an Illumina HiSeq 2000 system at the Beijing Genomics Institute (BGI).

## Plasmid constructions

Reporter DNA constructs were prepared on the basis of the pGL3R vector (*Stoneley et al., 1998*). Plasmids containing the 5′ leaders of test mRNAs were cloned between SpeI and NcoI. pGL3R-HCV and pRluc plasmids are described in *Andreev et al. (2009)*. The 5′ leader of *IFDR1* mRNA (NM_001550.3) was shortened by 38 nt at the 5′ end to correspond to the location of the likely predominant transcription start based on the analysis of available ESTs (Expression Sequence Tags) for this region. The 5′ leader of *PPP1R15B* (NM_032833.3) was extended at the 5′ end by 1 nt, which is present in the majority of available EST sequences.

For the same reasons, the 5′ leader of *UCP2* mRNA (NM_003355.2) was shortened by 16 nt and the 5′ leader of *PTP4A1* mRNA (NM_003463.4) by 547 nt; both 5′ leaders were cloned into pGL3R between Spe1 and Nco1. The 5′ leader of *SLC35A4* mRNA (NM_080670.2) was extended by 11 nt. To prepare pGL3-ATF4, the human *ATF4* 5′ leader was obtained by RT-PCR with primers GGGTAATACGACTCACTATAGGGTTTCTACTTTGCCCGCCCACAG and GGCGCCATGGTTGCGG TGCTTTGCTGGAATCG. The resulting product was digested with NcoI (underlined) and inserted into pGL3 vector at SmaI-NcoI sites.pGL3-ATF5 was prepared with the leader of ATF5 mRNA (NM_001193646.1) shortened by 62 nt.

The HCV-Fluc plasmid contained the T7 promoter, the entire viral 5′ leader, and the first 33 codons of viral ORF fused to Fluc and without its initiator codon and entire viral 3′ UTR.

Full length human ppp1R15A (GADD34) sequence fused with N-terminal FLAG tag was cloned in pcDNA 3.1 construct between HindIII and XbaI to prepare pcDNA GADD34 construct.

## mRNA preparation

mRNA preparation was carried out as described by *Dmitriev et al. (2007)*. Briefly, PCR products were obtained with a forward primer containing the T7 promoter (either the universal primer which anneals to the vector sequence immediately upstream of insertions, CGCCGTAATACGACTCACTATAGG GAGCTTATCGATACCGTCG or the T7 promoter-containing gene specific primer) and reverse primer containing an oligo(dT) stretch of 50 nt T50AACTTGTTTATTGCAGCTTATAATGG. To introduce stem loop structure, PCR products were obtained with forward primer containing the T7 promoter: CGCCGTAATACGACTCACTATAGGGAGTGGACTTCGGTCCACTCCCAGCTTATCGATACCGTCG.

To introduce the CAA$_6$ sequence upstream of the IFRD1 uORF, the following primer was used: CGCCGtaatacgactcactataGGGCAACAACAACAACAACAACAACATGTATCGTTTTCGATCACAGCTC.

The PCR products were then purified and used as templates for T7 RNA polymerase using in vitro RNA transcription by T7 RiboMAX Large Scale RNA Production kit (Promega, Fitchburg Center, WI). For preparation of m7G-capped transcripts the 3′-O-Me-m7GpppG (ARCA cap analogue, New England Biolabs, Ispwich, MA) was added to the transcription mix without GTP for 5 min to prime transcripts with cap followed by the addition of GTP (at a ratio of ARCA:GTP 10:1). The resulting RNAs were purified by LiCl precipitation and examined for integrity by PAGE.

## Cell culture, western blots and transfection procedures

Experiments with mRNA transfection were performed as described in *Andreev et al. (2012)*. Briefly, the mixture of 0.2 µg m$^7$G-capped Fluc mRNA and 0.01 µg m$^7$G-capped Rluc mRNAs per 1 well of 24-well plate was transfected to the cells at 70–80% confluency either with Lipofectamin 2000 (Invitrogen, Waltham, MA) or Unifectin 56 (Rusbiolink, Russian Federation). Simultaneously with transfection, cells were treated with either 40 µM sodium arsenite, 2.5 mM DTT or 250 nM Torin-1 (Torics Biosciences, Minneapolis, MN). Two hours later (or at the specified time interval), cells were harvested and luciferase activities were analysed with the Dual Luciferase Assay kit (Promega).

For experiments with GADD34 overexpression, cells were transfected with either pcDNA-GADD34 or control pcDNA3.1 one day prior to mRNA transfection. Plasmids were transfected with Fugene 6 (Promega) according to the manufacturer's instructions.

For western blotting, cells were rapidly lysed with buffer containing 1% SDS and 20 mM Tris–HCl pH 6.8 followed by brief sonication of the lysates. This was done to prevent post-translational modifications of proteins of interest during the lysis. Antibodies used in the study were: rabbit anti-EIF4EBP1 (AB3251; Chemicon International, Germany), rabbit anti-GAPDH (PTG10494-1-AP; Proteintech, Chicago, IL), rabbit anti-ATF4 (10835-1-AP; Proteintech), rabbit anti-phospho-p70 S6 kinase (Thr389) (9205S; Cell Signalling, Danvers, MA), rabbit anti-phospho-S6 ribosomal protein (Ser235/236) (2211S; Cell Signalling), rabbit anti-S6 ribosomal protein (2217S, Cell Signalling), rabbit anti-phospho eIF2 (S51) (SA-405; Enzo, New York, NY), and anti-FLAG-M2 (SigmaAldrich, St. Louis, MO). To remove non-specific binding, phospho-eIF2 antibodies (1:2500) were incubated along with 10% fetal bovine serum in TBS-T.

## Data analysis

Cutadapt (*Martin, 2011*) was used to remove the 3′ adapter of the reads (TGGAATTCTCGGGTGC CAAGG for the first replicate and CTGTAGGCACCATCAATAGATCGGAAGAGCACACGTCT GAACTCCAGTCAC for the second). Reads that did not map to either the 'spike-in' (ATGTACACG GAGTCGACCCGCAACGCGA) or rRNA sequences were aligned to the RefSeq catalogue (*Pruitt et al., 2014*) downloaded from the NCBI website on 15 August 2013. The alignment was carried out with Bowtie version 1.0.0 (*Langmead et al., 2009*), with parameters -a -m 100 –norc (all read mappings to the positive strand were taken with exception of those with more than 100 mappings). Reads that mapped to transcripts of more than one gene or multiple times to a transcript were discarded. In order to maximize the genuine ribosome footprints aligning to the transcriptome, ribo-seq reads with a length typical for monosomes (29–35 inclusive) were used for further analysis. In the case of multiple transcript variants, among the transcripts annotated as protein coding, the one with the highest ribo-seq read density in control conditions was brought forward for differential expression analysis.

The raw read count data were rescaled to normalize for the differences in the total number of reads mapped with a rescale factor F. For the first replicate the rescaled factor F for each sample was calculated as the difference by which the total number of mapped reads exceeds the lowest total number of mapped reads out of two conditions, that is:

$$F_n = \frac{\sum_i x_{ni}}{\min_i x_{ni}},$$

where $x_{ni}$ is the number of mapped reads from sample n in the condition i. This normalization was carried out independently for ribo-seq and mRNA-seq.

The second replicate was sequenced on Illumina Miseq and Hiseq 2000 instruments and obtained sequence reads were aggregated. The number of 'spike-in' reads was used to rescale the read counts

with a similar approach as for replicate 1. The raw read count of each sample was divided by the rescaling factor $F$ calculated as above with the only difference that $x_{ni}$ represents the number of 'spike-in' reads from sample $n$ in the condition $i$. This rescaling was also implemented for the ribosomal profiles of individual transcripts shown in *Figure 2*.

The normalized read counts of ribo-seq reads aligning to the coding regions (as determined by inferred locations of the A-site codons) and of mRNA-seq reads aligning to the entire transcript were used for the differential expression analysis. For ribo-seq reads the A-site codon of the elongating ribosome was inferred to be the 17th or 18th nt of the read from its 5′ end depending on the read length.

Transcripts were binned based on the number of mapped reads (expression/coverage level) in one of the conditions where this value is the minimal. For the analysis of differential translation efficiency the minimum value (referred to as the minimum expression level) was taken from four conditions while, for the analysis of differential RNA level, only RNA-seq reads obtained under control and stress conditions were used. With the minimum expression level threshold of two reads, transcripts were sorted in ascending order and arranged in bins of size 300. Each bin had transcripts with a similar number of mapped reads and was analysed independently. The mean and standard deviation of change in expression of the transcripts within each bin was used to determine a Z-score for each transcript. For the remaining transcripts of insufficient number to be binned (<300), the mean and standard deviation was obtained from the previous bin. The Z-score determined for each transcript enabled comparison between bins.

The analysis of translation of mRNA leaders was carried out for the transcripts with at least two normalized read counts in each of all four experiments/conditions. An uORF was defined as a sequence of sense codons uninterrupted with a stop codon and beginning with an AUG codon located upstream of the annotated CDS. In the case of uORFs overlapping CDS, the 5′ end of CDS was considered as the end of the uORF in order to avoid ambiguity in assigning ribo-seq reads to one of the two overlapping ORFs. Nested uORFs (those contained within uORFs in the same frame) were excluded for the same reason. The TE of an uORF was estimated as the average density of ribo-seq reads in the uORF divided by the average density of the mRNA-seq reads for the corresponding mRNA. An uORF was considered to be translated if more than five ribo-seq reads aligned to it. For transcripts with more than one translated uORF, the properties of the uORF with the highest number of aligned ribo-seq reads were used.

For the purpose of the analysis represented in *Figure 3C*, the centre of ribosome density was defined as the minimal mRNA coordinate for which the number of ribo-seq reads aligning 5′ of the corresponding location is the same or greater than the number of ribo-seq reads aligning 3′ of the corresponding location. This value was determined for genes under arsenite and control conditions. The difference in ribosome footprint density was divided by the CDS length to prevent skewing of results in favour of transcripts with longer coding regions.

The list of human cellular IRES was obtained from IRESite (*Mokrejs et al., 2010*) on 2 August 2014.

We identified the most probable translation initiation sites by manually examining the ribo-seq profiles of eight translationally resistant genes (*PPP1R15A*, *IFRD1*, *SLC35A4*, *C19ORF48*, *PTP4A1*, *PCNXL4*, *UCP2*, *PPP1R15B*). *ATF4* and *ATF5* were not included as these appeared to be regulated by an alternative method. uORF initiation sites with an AUG or CUG were selected based on their ability to fit with the observed profile upon manual examination. AUG codons were preferred, but the surrounding consensus sequence was not considered. A sequence logo of the initiation sites (−4 to +3) was produced with WebLogo (*Crooks et al., 2004*).

For *Figure 3—figure supplement 1B*, the sequences of all coding transcripts were included to determine the frequency of the initiation sites for both annotated start sites and for AUG sites in the leaders. The relative frequency was obtained by dividing the number of occurrences of a particular sequence by the total number. For example, the sequence GGCCATGG, the most common Kozak sequence in CDS sites occurring 640 times in 35,851 transcripts, has a relative frequency of $(640/35,841) \times 100 = 1.78\%$.

The free energy of leaders was estimated with RNAfold (*Lorenz et al., 2011*). The first 240 nt was used (transcripts with shorter leaders were excluded) as free energy of RNA is related to its length. We chose 240 nt as this was the length of the shortest leader of resistant mRNAs with a translated uORF.

## Data access

Sequences of ribosome profiling libraries have been deposited into the NCBI Gene Expression Omnibus portal under the accession number GSE55195.

## Acknowledgements

We are grateful to Gary Loughran (UCC) for the critical reading of the manuscript and useful comments. We also thank Irwin Jungreis (MIT), Mike Lin (DNAnexus), and Manolis Kellis (MIT) for permitting the use of CodAlignView (Jungreis I, Lin M, Kellis M. CodAlignView: a tool for visualizing protein-coding constraint) for figure preparations.

## Additional information

### Funding

| Funder | Grant reference number | Author |
| --- | --- | --- |
| Wellcome Trust | 094423 | Pavel V Baranov |
| Science Foundation Ireland (SFI) | 12/IA/1335 | Pavel V Baranov |
| Russian Foundation for Basic Research (RFBR) | 12-04-32039 | Dmitry E Andreev |
| Russian Foundation for Basic Research (RFBR) | 12-04-33196 | Sergey E Dmitriev |
| Russian Science Foundation | 14-14-00127 | Dmitry E Andreev |
| European Molecular Biology Organization (EMBO) | | Dmitry E Andreev |

The funders had no role in study design, data collection and interpretation, or the decision to submit the work for publication.

### Author contributions

DEA, Conception and design, Acquisition of data, Analysis and interpretation of data, Drafting or revising the article; PBFO'C, Analysis and interpretation of data, Drafting or revising the article; CF, EMK, IMT, PC, DWM, Acquisition of data, Drafting or revising the article; SED, Conception and design, Acquisition of data, Drafting or revising the article; INS, Conception and design, Drafting or revising the article; PVB, Conception and design, Analysis and interpretation of data, Drafting or revising the article

### Author ORCIDs

Patrick BF O'Connor,  http://orcid.org/0000-0003-1085-2795
Sergey E Dmitriev,  http://orcid.org/0000-0002-1774-8475
Pavel V Baranov,  http://orcid.org/0000-0001-9017-0270

## Additional files

### Major datasets

The following dataset was generated:

| Author(s) | Year | Dataset title | Dataset ID and/or URL | Database, license, and accessibility information |
| --- | --- | --- | --- | --- |
| Andreev DE, O'Connor PB, Fahey C, Kenny EM, Terenin IM, Dmitriev SE, Cormican P, Morris DW, Shatsky IN, Baranov PV | 2014 | Ribosome profiling data obtained from HEK293T cells 30 min after treatment with arsenite to a final concentration of 40 μM | http://www.ncbi.nlm.nih.gov/geo/query/acc.cgi?acc=GSE55195 | Publicly available at NCBI Gene Expression Omnibus. |

The following previously published dataset was used:

| Author(s) | Year | Dataset title | Dataset ID and/or URL | Database, license, and accessibility information |
| --- | --- | --- | --- | --- |
| N/A, | 2013 | Homo sapiens SLC35A4 gene for alternative protein SLC35A4, clone 9074 | http://www.ebi.ac.uk/ena/data/view/HF548106 | Publicly available at European Nucleotide Archive. |

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
