## [Decision Letter]

Thank you for sending your work entitled “mRNAs with efficiently translated uORFs
resist stress-induced translation suppression inflicted by eIF2 phosphorylation”
for consideration at *eLife*. Your article has been favorably evaluated
by James Manley (Senior editor) and three reviewers, one of whom is a member of our
Board of Reviewing Editors.

The Reviewing editor and the other reviewers discussed their comments before we reached
this decision, and the Reviewing editor has assembled the following comments to help you
prepare a revised submission.

In this study, you identify mRNAs that continue to be translated in the presence of
sodium arsenite. Ribosome profiling of HEK293T under normal and stress conditions,
identified a small number of mRNA whose translation is resistant to eIF2-alpha
phosphorylation. Interestingly, the majority of these mRNAs contain uORFs that are
efficiently translated under normal conditions. Experiments done with wild type and
mutant mRNA reporters demonstrated the importance of uORF for translational
resistance.

The reviewers thought that in general the work is interesting. It incorporates an
important technology to assess translational control genome wide. The genome-wide
analysis is the most important contribution of the manuscript and there are several new
genes identified whose translational regulation could be important for cellular
adaptation to arsenite stress. Also, besides identifying several new stress-resistant
mRNAs, you describe a specific characteristic that is associated with these mRNAs, an
efficiently translated uORF.

Two of the reviewers noted that a mechanistic explanation of how uORFs provide
translational resistance to stress is lacking, however.

In addition, the work does not break new ground in our understanding of translational
control during stress and eIF2 phosphorylation. No new mechanisms of translational
control were described, but rather the study provides a broader picture supporting the
idea that certain uORFs are central to mechanisms facilitating translation during
stress.

Because the results are novel and important we are prepared to consider a revised
version that will address some important questions raised by the reviewers as
follows:

1) Given the small number of eIF2-alpha phosphorylation resistant transcripts, it is
important to validate them by RT-qPCR/polysome profiling. In addition, it is rather
surprising that the list of stress-resistant mRNAs does not include mRNAs that were
reported to possess IRESs. This should be discussed.

2) The manuscript should be more precise in explaining the calculation that was used to
classify gene transcripts that are resistant to arsenite stress. Classification of
resistant mRNAs could be based on either the change in RNAseq reads or the Z score. This
clarification is important because mRNAs that have a positive Z score may still
experience a reduction in translation during arsenite treatment, in which case the Z
score may not be an the best measure of mRNA resistance. Explain and justify which
calculation was used to classify mRNAs as being resistant and provide a full description
of what delineates translation resistance from repression during arsenite stress.

3) The manuscript should delineate preferential translation from translation resistance
(sometimes called tolerance). In preferential translation there is poor mRNA translation
during non-stressed conditions and high levels of translation in response to stress and
eIF2 phosphorylation. The literature indicates that ATF4 is an example of preferential
translation in response to eIF2 phosphorylation as supported by Figure 1. Curiously in this manuscript, arsenite stress did not
induce the expression of the luciferase reporter with the 5'-leader of the ATF4
transcript (Figure 4) and there was perhaps only
a modest enhancement of ribosome occupancy in the mRNA CDS during arsenite stress as
judged by ribosome profiling. Perhaps arsenite differs from other stress conditions that
were reported to induce preferential translation of ATF4 mRNA (e.g. pharmacological
inducers of ER stress).

4) *SLC35A4* appears to be the only gene that showed strong enhanced
translation in response to arsenite translation (i.e. preferential translation). Does
the 5'-leader of the *SLC35A4* gene transcript confer enhanced
expression in a reporter assay during arsenite stress?

5) Do the uORFs in the resistant gene transcripts display any differences from those
whose translation was repressed during arsenite stress?

6) The manuscript refers to eIF2 phosphorylation and translational control, as
highlighted in the title, but there were no experiments establishing cause/effect
between eIF2 phosphorylation and the translation expression of a gene. Rather, it was
only inferred based on translational control mechanisms previously described in the
literature for ATF4 and the related ISR genes.

7) Could you include analysis of mRNAs either for the context around the AUG in the uORF
or whether non-AUG codons might be used (if in a good context)? This would require
additional data mining but no additional experiments. This might also allow for an
explanation of the 11 AUGs in the *SLC35A4* mRNA that do not seem to
serve as initiation codons.

8) A major concern is the sampling time. From Figure 1 it appears that the hyper-phosphorylation of 4E–BP1 disappears at 1
hour (although the gel is not of high enough quality to tell; there is the possibility
of smiling of the gel bands). The GAPDH loading control is overexposed and, therefore,
inadequate. The phosphorylation of eIF2alpha is up at 0.5 hours but maximal at 2 hours.
The concern is the possible crossover of 4E–BP1 activation (loss of
phosphorylation) and the onset of eIF2alpha phosphorylation. The authors should more
rigorously exclude indirect targeting of mTORC1–4EBP and mTORC1–S6K
pathways of protein synthesis regulation by arsenite treatment. The 4EBP1 blot on Figure 1 should be less exposed to make the bands
more discernible. The blots showing no change in p-4EBP1 and p-S6K levels under their
conditions of arsenite treatment should also be shown. Do the authors have an
independent control for determining the level of cap-dependent translation vs. the level
of eIF2-dependent translation?

[Editors' note: further revisions were requested prior to acceptance, as
described below.]

Thank you for resubmitting your work entitled “Translation of leaders is
pervasive in genes resistant to eIF2 repression” for further consideration at
*eLife*. Your revised article has been evaluated by James Manley
(Senior editor), a member of the Board of Reviewing Editors, and two reviewers. The
manuscript has been improved but there are some significant remaining issues that need
to be addressed before proceeding, as outlined below.

There are two remaining major concerns:

The first concern was the bioinformatics and reluctance to incorporate fold-change into
the core statistics of the manuscript. This is a shortcoming and confuses interpretation
of the identification of genes whose translation is induced upon arsenite stress. This
is the potential novelty of the manuscript.

The other major concern was the experimental validation of the ribosome profiling. The
reporter assays do not fully support the profiling data, and the controls, e.g. ATF4,
did not follow the expected induced translation with increased eIF2 phosphorylation. In
the rebuttal letter (the Methods of the manuscript were not detailed on this point), the
reporter mRNA was stated to be transfected in cultured cells for 1 hour prior to
arsenite treatment for an additional 2 hours (the Methods section stated the later
treatment time). There is a concern that the non-stressed cells are not truly
non-stressed as there is likely to be insufficient recovery from the transfection using
lipofectamin 2000. This could explain why there were virtually no genes that were
preferentially translated in the reporter assay, just different degrees of repression.
The non-stressed conditions were not without the underlying membrane stress of the
transfection agent. Also 40 micromolar is a large amount of arsenite to stress the
cells.

Below is the description of the concerns in more detail:

The first is centered on reviewer concern number 3 that indicated that there should be
clear delineation between the levels of translational control, e.g. preferential
translation and translation resistance (or tolerance). In the first paragraph of the
response to reviewer concern 3 the authors provide a definition of preferential
translation as an increase of translation with a high Z-score and a fold-change
>1 and tolerant with a high Z-score and fold change as <1. This was a
straightforward conceptual framework involving Z-score and fold-change that should be
incorporated into the manuscript analysis and ranking.

The second concern is how the results from ribosomal profiling analysis compare to other
approaches previously used in the literature. This is the heart of the manuscript. In
Reviewer concern 1, the response stated that qPCR and polysome analysis would not be
appropriate for this analysis despite its utility in the literature. As a consequence,
the manuscript elected to rely on reporter assays involving transfections of mRNAs
featuring 5'-leaders of targeted gene transcripts upstream of the firefly
luciferase coding sequence. ATF4, a well-characterized preferentially translated gene
transcript was used as a positive control, and the artificial gene transcript pGL3 was
an example of a gene that is repressed by eIF2 phosphorylation and stress. Curiously,
ATF4 translation was not preferentially translated, but displayed a partial resistance
to arsenite stress. The authors’ argument for why ATF4 ribosome profiling data
and reporter data are in agreement (even though the ATF4 reporter is not induced) is
based on “technical limits of methods accuracy”. The details of the
reporter assay were not fully clear in the original or present manuscript, but the
response letter provided additional details in the response to reviewer concern number
3. The mRNAs were transfected into HEK293T cells and 1 hour after transfection the cells
were treated with arsenite for 2 hours prior to harvesting and analysis. There are some
concerns about the timing in this protocol, such as would 1 hour be sufficient time for
the resolution of the membrane stress triggered by the transfection protocol?
Furthermore, it is suggested that even though the *SLC35A4* reporter was
not inducible during arsenite stress, that this is not in disagreement with the ribosome
profiling dataset because *SLC35A4* mRNA levels are low in the profiling
dataset. As the purpose of using the Z-score was to eliminate variability and error in
the assessment of the ribosome profiling dataset, is this sufficient justification for
why the two assay results do not appear to be consistent? In a related point in concern
number 6, it was suggested that the manuscript should focus on “establishing
cause/effect between eIF2 phosphorylation and translation expression of a gene.”
This could be directly addressed in the reporter assay, and the explanation involving an
alternative stress (DTT) and mTOR did not appear to adequately address this basic
concern.

Other comments:

1) The English is still a bit rough, but not to the point where the reader cannot
understand what is intended.

2) Although in the position of being a “break out” paper, it would be nice
if there was some effort expended at a possible explanation although it would be
obviously speculative (see below). To just leave the paper to end on “there a
black box that allows for this resistance to eIF2 repression” is unsatisfying
(although accurate).

3) The authors stress the resistance to eIF2 repression, but do not stress as well that
most of the proteins are poorly expressed. Using the number of reads scale in Figure 2 where ATF4 is 43, the other proteins fall
mostly into the 0 to 2-8 range. This inefficiency is also reflected in the reporter
assays in Figure 4 where pGL3 is 100%, the
remaining normal configuration for most constructs yields expression levels in the 5 to
10% of pGL3 range.

4) That said, the low level of expression may be entirely appropriate for phosphatases,
kinases, transcription factors, etc. that are not required in large amounts.

---

## [Author Response]

*Editorial comment*:

*When you submit the revised version of your manuscript, we would also like you
to revise the title to make it more broadly understandable and more readable. The
presence of three acronyms in the present title, combined with its length, reduce the
readability of the title so, if possible, we would be grateful if you could reduce
the number of acronyms and/or length of the title*.

We revised the title of the manuscript to a shorter variant and reduced the number of
acronyms. The new title is: “Translation of leaders is pervasive in genes
resistant to eIF2 repression”.

*Two of the reviewers noted that a mechanistic explanation of how uORFs provide
translational resistance to stress is lacking, however*.

*In addition, the work does not break new ground in our understanding of
translational control during stress and eIF2 phosphorylation. No new mechanisms of
translational control were described, but rather the study provides a broader picture
supporting the idea that certain uORFs are central to mechanisms facilitating
translation during stress*.

*Because the results are novel and important we are prepared to consider a
revised version that will address some important questions raised by the reviewers as
follows*:

*1) Given the small number of eIF2-alpha phosphorylation resistant transcripts,
it is important to validate them by RT-qPCR/polysome profiling. In addition, it is
rather surprising that the list of stress-resistant mRNAs does not include mRNAs that
were reported to possess IRESs. This should be discussed*.

We agree that validation of the predictions using an alternative method is important.
However, we have concerns whether RT-qPCR of polysome fractions would be applicable to
our case. For most stress-resistant mRNAs identified in our study, the translational
response is accompanied with a significant reduction of ribosome occupancy at their
uORFs. Therefore, these mRNAs may shift to lighter polysome fractions even if
translation of the main coding region is increased. While ribosome profiling provides
information on local changes of ribosome density within the same mRNA, polysome
profiling cannot distinguish translation of the main coding region from translation of a
uORF.

Therefore, in order to validate ions we chose to test the ability of 5’ leaders
to provide resistance using mRNA constructs with reporter genes. We created three new
reporters and demonstrated that in addition to PPP1R15B, and *IFRD1*
(tested earlier) leaders of UCP2, *SLC35A4* and PTP4A1 are sufficient for
stress-resistant translation (Figure 4—figure supplement 2).

The phenomenon of internal ribosome entry site (IRES) initiation has been known for
decades since its discovery in EMCV and Poliovirus. However, only a few examples of
IRESes were reported to be independent of eIF2 inactivation and to be able to operate in
eIF2 independent mode. We extracted the list of human mRNAs (reported to contain IRES)
from IRESite and address the question whether these mRNAs are resistant to arsenite
treatment. We did not find any of these mRNAs to be resistant to arsenite treatment.
However, not all of these mRNAs are expressed at a level sufficient for detecting
resistance. We discuss this now in the manuscript, we also made a new table (Table 1) that lists mRNAs that were reported to
have IRES.

*2) The manuscript should be more precise in explaining the calculation that was
used to classify gene transcripts that are resistant to arsenite stress.
Classification of resistant mRNAs could be based on either the change in RNAseq reads
or the Z score. This clarification is important because mRNAs that have a positive Z
score may still experience a reduction in translation during arsenite treatment, in
which case the Z score may not be an the best measure of mRNA resistance. Explain and
justify which calculation was used to classify mRNAs as being resistant and provide a
full description of what delineates translation resistance from repression during
arsenite stress*.

Z-score transformation is widely used for differential gene expression analysis, because
it provides a simple method for comparing changes of gene expression across genes
expressed at different levels. This cannot be done with fold changes because the same
fold change may be statistically significant for a highly expressed gene, but not for a
low expressed gene. This is because measurements for low expressed genes are expected to
have higher variability and lower accuracy due to poorer sequencing coverage. This is
clearly seen in scatter plots (Figure 1—figure supplement 1) where the distributions of read counts have higher variability
for genes with low read count.

Nonetheless, we agree that the information on absolute changes of expression is also
important and therefore we provide both. However, our ranking of resistance is based on
the Z-score and we now describe this in greater detail in the manuscript. We also
provide a new reference to a seminal review that describes the topic in further detail:
[55], Nat Genet, 2002,
32:496-501 PMID: 12454644.

It is true that genes experiencing a reduction of translation may have a high positive
Z-score. However, a high Z-score would indicate that this reduction is exceptionally low
in comparison with the reduction experienced by the rest of genes. This indicates
resistance. On the contrary a high fold change does not necessarily indicate the
resistance. If the Z-score magnitude is low, it means that such change (no matter how
high) may occur between two replicas produced under the same conditions and therefore
such a change may not have biological meaning. A low Z-score magnitude does not
necessarily mean that the corresponding mRNA is not resistant. It means that we cannot
detect it, either because the mRNA is not resistant or because the level of its
expression is insufficient for resistance detection.

*3) The manuscript should delineate preferential translation from translation
resistance (sometimes called tolerance). In preferential translation there is poor
mRNA translation during non-stressed conditions and high levels of translation in
response to stress and eIF2 phosphorylation. The literature indicates that ATF4 is an
example of preferential translation in response to eIF2 phosphorylation as supported
by*
Figure 1*. Curiously in
this manuscript, arsenite stress did not induce the expression of the luciferase
reporter with the 5'-leader of the ATF4 transcript (*Figure 4*) and there was
perhaps only a modest enhancement of ribosome occupancy in the mRNA CDS during
arsenite stress as judged by ribosome profiling. Perhaps arsenite differs from other
stress conditions that were reported to induce preferential translation of ATF4 mRNA
(e.g. pharmacological inducers of ER stress)*.

In order to delineate “preferential translation” from
“tolerance” in a high-throughput study we need to establish a quantitative
definition of “preferential translation” and “tolerance”. If
we define “preferential translation” as an increase of translation and
“resistance” as a weak decrease in translation, then genes with a high
Z-score and a fold change >1 should be classified as “preferentially
translated” and those with high a Z-score and a fold change <1 as
“tolerant”.

Based on this definition, ATF4 is preferentially translated in our conditions. Western
blotting (Figure 1) shows increase of ATF4
protein levels. The actual level of increase is likely to be an overestimate because
under normal conditions ATF4 is hydroxylated and rapidly degrades (see Koditz et al.,
2007, Blood 110:3610-7). Stabilization of ATF4 under stress conditions may lead to its
accumulation irrespective of a change in translation. Ribosome profiling (Figures 1 and 2 as well as Supplementary
Table) also indicates increase of ATF4 mRNA translation by ∼30%. The reporter
assay in HEK293 cells (Figure 4) shows almost no
change. The reporter assay in Huh7 cells (Supplementary Figure S3B) shows about 25%
increase of translation. Given the technical limits of the accuracy of the methods (note
the error bar for the reporter assays), the data are in agreement. Importantly, we
report normalized absolute Fluc values rather than Fluc/Rluc values (which is more
common practice). In our case upon arsenite treatment Fluc/Rluc values significantly
increase due to strong inhibition of Rluc translation.

Therefore, while it is possible and even likely that the stress response to arsenite
treatment differs from the response to other stresses inducing ATF4 translation, we
cannot draw such a conclusion solely based on ATF4 behavior. Relevant to this, in
experiments not described in the manuscript we also observed variation in ATF4
translation response depending on the severity of the stress (concentration of arsenite
and duration of treatment). The translation of a particular ORF under stress condition
is modified by the global reduction of translation and an individual mechanism of
translation activation. It can be expressed as TE(stress)=[TE(normal)Ka]/Ki,
where Ka represents mRNA-specific activation under given condition and Ki represents
global inhibition. When Ka>Ki, TE(stress)>TE(normal) and the mRNA is
preferentially translated. When Ka<Ki, TE(stress)<TE(normal) and the mRNA
is stress tolerant. Ka/Ki is unlikely to be a constant, thus the same mRNA could be
tolerant or preferentially translated depending on conditions. To illustrate, consider
the extreme points: under very low concentrations of eIF2 phosphorylation no change is
likely; under very high concentrations, all eIF2 dependent translation will cease,
including for resistant genes. Between these points there may be specific conditions
under which translation of a specific mRNA is higher than that under normal conditions
(preferentially translated), but it certainly would not be so at the entire range of
phosphorylated eIF2 concentrations.

We decided to introduce an alteration to our reporter assay in comparison with the
original to make it more relevant to address the effect on ongoing translation. In the
original work mRNAs were transfected into the cells simultaneously with arsenite
treatment. In the modified assay, we treated cells with either arsenite or cycloheximide
an hour after transfection. While cycloheximide blocks all translation, arsenite
treatment permits translation of reporter mRNAs with leaders from resistant mRNAs.
Interestingly, for *SLC35A4* mRNA, arsenite treatment results in the
production of 15 times more luciferase than that for cycloheximide treatment.

*4)* SLC35A4 *appears to be the only gene that showed strong
enhanced translation in response to arsenite translation (i.e. preferential
translation). Does the 5'-leader of the* SLC35A4 *gene
transcript confer enhanced expression in a reporter assay during arsenite
stress?*

For the revised version of the manuscript we designed a reporter construct with
*SLC35A4* 5’ leader. The reporter assay demonstrated only
moderate upregulation of the main ORF translation (Figure 4—figure supplement 1 and Figure 4—figure supplement 2). However,
*SLC35A4* resistance was still the strongest in comparison with other
tested 5’ leaders. Although the gene was found to have an apparently large
translational increase detected with ribosome profiling it was associated with a large
degree of uncertainty due to its very low expression as explained in our answer to the
comment 2. Thus the results are not conflicting.

5) Do the uORFs in the resistant gene transcripts display any differences from
those whose translation was repressed during arsenite stress?

The organization of uORFs (in terms of their numbers and length) varies considerably
among 5’ leaders of all resistant mRNAs as can be seen in Figure 2. However, the presence of at least one comparatively long
uORF (>20 codons) is a common feature (see Figure 3).

To compare 5’ leaders of resistant and repressed mRNAs we analyzed the nucleotide
context of uORFs initiation starts (Kozak) and secondary structure potential within
5’ leaders. We have not found significant difference in the distribution of Kozak
contexts (Figure 3—figure supplement 1).
Also see our response to the comment 7 below.

We found that the resistant mRNAs have lower secondary structure potential (based on
free energy estimates of putative RNA structures) within the first 240 nt of the leaders
in comparison with an average for non-resistant mRNAs. However, many non-resistant mRNAs
have even lower RNA secondary structure potential (Figure 3—figure supplement 1).

*6) The manuscript refers to eIF2 phosphorylation and translational control, as
highlighted in the title, but there were no experiments establishing cause/effect
between eIF2 phosphorylation and the translation expression of a gene. Rather, it was
only inferred based on translational control mechanisms previously described in the
literature for ATF4 and the related ISR genes*.

We believe that the comment can be formulated as the following. There are no experiments
demonstrating that the translation control observed in response to the arsenite
treatment is mediated via eIF2 phosphorylation.

To demonstrate that translational control in response to arsenite treatment is mediated
through eIF2 phosphorylation, it is necessary to show that the treatment results in eIF2
phosphorylation and that it does not affect translation by other means. The first part,
the causative link between arsenite treatment and eIF2 phosphorylation has been
established before (see [42],
JBC, 280: 16925-33, PMID: 15684421. To address the second part we used two positive
controls, HCV and ATF4. HCV can operate in eIF2 independent mode. Its translation was
not inhibited by arsenite treatment suggesting that translation elongation was not
affected (Figure 4). As noted by the referee ATF4
was also a control whose translation control mechanism was established previously.

To rule out other pathways affecting cap-dependent initiation we tested whether
5’ leaders of arsenite resistant mRNAs also resistant to mTOR inhibition and
tested the reports in the presence of torin-1. No differential suppression was observed
(Figure 4—figure supplement 3).
Finally, in order to eliminate a possibility of a translational response to arsenite
treatment instigated by means other than eIF2 phosphorylation we analyzed expression of
the reporters in the presence of DTT, another well-known inducer of eIF2 phosphorylation
(see [53], MCB, 127-128:255-65,
PMID: 7935356). The results were similar to those observed under the arsenite treatment
(Figure 4—figure supplement 1). It
is, of course, unlikely that arsenite treatment effects eIF2 exclusively. However, based
on our data, we believe that the major effect on translation is due to eIF2
phosphorylation.

*7) Could you include analysis of mRNAs either for the context around the AUG in
the uORF or whether non-AUG codons might be used (if in a good context)? This would
require additional data mining but no additional experiments. This might also allow
for an explanation of the 11 AUGs in the* SLC35A4 *mRNA that do not
seem to serve as initiation codons*.

Thank you for this suggestion. We have included such analysis in the revised version of
the manuscript (Figure 3—figure supplement 1). Surprisingly not all of the translated uORFs contain initiation codons in
good context.

We also checked how the Kozak context of uORF in 5’ leader of
*IFRD1* affects the resistance. The wild type uAUG in its leader is in
suboptimal Kozak context (-3A but +4U). The +4U/G substitution slightly
inhibited the translation of the reporter, most likely due to increased translation of
the uORF, but it did not alter the reporter response to the stress (Figure 4—figure supplement 3).

*8) A major concern is the sampling time. From*
Figure 1
*it appears that the hyper-phosphorylation of 4E–BP1 disappears at 1 hour
(although the gel is not of high enough quality to tell; there is the possibility of
smiling of the gel bands). The GAPDH loading control is overexposed and, therefore,
inadequate. The phosphorylation of eIF2alpha is up at 0.5 hours but maximal at 2
hours. The concern is the possible crossover of 4E–BP activation (loss of
phosphorylation) and the onset of eIF2alpha phosphorylation. The authors should more
rigorously exclude indirect targeting of mTORC1–4EBP and mTORC1–S6K
pathways of protein synthesis regulation by arsenite treatment. The 4EBP1 blot
on*
Figure 1
*should be less exposed to make the bands more discernible. The blots showing no
change in p-4EBP1 and p-S6K levels under their conditions of arsenite treatment
should also be shown. Do the authors have an independent control for determining the
level of cap-dependent translation vs. the level of eIF2-dependent
translation?*

Thank you for pointing this out. We substituted the western blot in Figure 1 with one of a better quality and a lower exposure.
Further we added an additional line with the same control lysate (last line) to ensure
the absence of gel artifacts. We did not detect any significant changes in 4E–BP1
dephosphorylation after 30 min of 40 µM arsenite treatment (compare lanes 1 and
2). We did not detect changes in the levels of phosphorylated p70 S6k and phosphorylated
rpS6 during such treatment. We added the western gel to Figure 1 panel.

We also addressed this question by analyzing how mTOR sensitive mRNAs responded to the
stress using our ribosome profiling dataset. We mapped mRNAs which were shown to be
downregulated upon treatment with mTOR inhibitor PP242 ([25], Nature, 485:55-61 PMID: 22367541). Their
translational efficiency decrease upon arsenite treatment was ∼25% greater than
the average (6.54-fold v 5.2-fold), see (Figure 1—figure supplement 2). Thus, while it is possible that arsenite
treatment effects mTOR pathway, its impact would be subsidiary to the eIF2 pathway.
Interestingly, there are several outliers like PABPC1 and RPL12 (Figure 1—figure supplement 2, left bottom corner) which
behaves differently from the rest of mTOR sensitive mRNAs and which are exceptionally
sensitive to the arsenite treatment. Perhaps, these mRNAs are sensitive to both types of
stress pathways.

[Editors' note: further revisions were requested prior to acceptance, as
described below.]

*There are two remaining major concerns*:

*The first concern was the bioinformatics and reluctance to incorporate
fold-change into the core statistics of the manuscript. This is a shortcoming and
confuses interpretation of the identification of genes whose translation is induced
upon arsenite stress. This is the potential novelty of the manuscript*.

We feel that this concern, at least in part, is a result of misunderstanding. The
fold-change statistics was incorporated in both, the original and the revised version of
the manuscript. Fold changes of RNAseq, riboseq and TE signals are given for every gene
in the source data file 1. They are provided for selected genes in Figure 1 and are shown in axis y in scatter plots in Figure 1. Perhaps, the confusion came from the
rebuttal letter where we overzealously explained a need to estimate statistical
significance of change folds using Z-score. We apologise for the confusion caused.

*The other major concern was the experimental validation of the ribosome
profiling. The reporter assays do not fully support the profiling data, and the
controls, e.g. ATF4, did not follow the expected induced translation with increased
eIF2 phosphorylation. In the rebuttal letter (the Methods of the manuscript were not
detailed on this point), the reporter mRNA was stated to be transfected in cultured
cells for 1 hour prior to arsenite treatment for an additional 2 hours (the Methods
section stated the later treatment time). There is a concern that the non-stressed
cells are not truly non-stressed as there is likely to be insufficient recovery from
the transfection using lipofectamin 2000. This could explain why there were virtually
no genes that were preferentially translated in the reporter assay, just different
degrees of repression. The non-stressed conditions were not without the underlying
membrane stress of the transfection agent. Also 40 micromolar is a large amount of
arsenite to stress the cells*.

We considered the possibility that transfection, as any other manipulation with cultured
cells, may stress the cells inducing eIF2 phosphorylation. We conducted a new experiment
that we now describe in the manuscript (also see response further below). In this
experiment we overexpressed *ppp1r15a* encoding GADD34 for one day prior
to stress induction. GADD34 should dephosphorylate eIF2 if it occurs due to the membrane
stress. However, we did not observe significant increases in reporter assays in
comparison with cells where GADD34 was not overexpressed, see Figure 4—figure supplement 4. Therefore the stress induced
by transfection, if occur, is unlikely to be significant.

In addition, it was demonstrated earlier by Anderson and coauthors that, unlike
nucleofection, lipofection does not induce eIF2 phosphorylation (see Figure 2 in BR Anderson et al., 2013, Gene Therapy,
PMID: 22301437).

We agree that it is possible that 40 uM concentration of arsenite is too high for
obtaining the conditions for maximum translation of ATF4 mRNA. We optimized the arsenite
concentration and duration of the stress to generate conditions under which global
translation would be substantially but not completely inhibited as can be judged from
the polysome profiles. We believe that such conditions would be the most informative for
the ribosome profiling experiment.

*Below is the description of the concerns in more detail*:

*The first is centered on reviewer concern number 3 that indicated that there
should be clear delineation between the levels of translational control, e.g.
preferential translation and translation resistance (or tolerance). In the first
paragraph of the response to reviewer concern 3 the authors provide a definition of
preferential translation as an increase of translation with a high Z-score and a
fold-change >1 and tolerant with a high Z-score and fold change as <1.
This was a straightforward conceptual framework involving Z-score and fold-change
that should be incorporated into the manuscript analysis and ranking*.

We agree that such a classification could be useful and incorporated it in the new
version of the manuscript (see the beginning of the subsection “Efficient
translation of uORFs combined with inefficient translation of CDS is a predictor of
stress resistant mRNAs”).

*The second concern is how the results from ribosomal profiling analysis compare
to other approaches previously used in the literature. This is the heart of the
manuscript. In Reviewer concern 1, the response stated that qPCR and polysome
analysis would not be appropriate for this analysis despite its utility in the
literature. As a consequence, the manuscript elected to rely on reporter assays
involving transfections of mRNAs featuring 5'-leaders of targeted gene
transcripts upstream of the firefly luciferase coding sequence. ATF4, a
well-characterized preferentially translated gene transcript was used as a positive
control, and the artificial gene transcript pGL3 was an example of a gene that is
repressed by eIF2 phosphorylation and stress. Curiously, ATF4 translation was not
preferentially translated, but displayed a partial resistance to arsenite stress. The
authors’ argument for why ATF4 ribosome profiling data and reporter data are
in agreement (even though the ATF4 reporter is not induced) is based on
“technical limits of methods accuracy”. The details of the reporter
assay were not fully clear in the original or present manuscript, but the response
letter provided additional details in the response to reviewer concern number 3. The
mRNAs were transfected into HEK293T cells and 1 hour after transfection the cells
were treated with arsenite for 2 hours prior to harvesting and analysis. There are
some concerns about the timing in this protocol, such as would 1 hour be sufficient
time for the resolution of the membrane stress triggered by the transfection
protocol? Furthermore, it is suggested that even though the SLC35A4 reporter was not
inducible during arsenite stress, that this is not in disagreement with the ribosome
profiling dataset because SLC35A4 mRNA levels are low in the profiling dataset. As
the purpose of using the Z-score was to eliminate variability and error in the
assessment of the ribosome profiling dataset, is this sufficient justification for
why the two assay results do not appear to be consistent? In a related point in
concern number 6, it was suggested that the manuscript should focus on
“establishing cause/effect between eIF2 phosphorylation and translation
expression of a gene.” This could be directly addressed in the reporter assay,
and the explanation involving an alternative stress (DTT) and mTOR did not appear to
adequately address this basic concern*.

Thank you for the useful comments and suggestions.

1) We now addressed the “cause/effect” concern by conducting the following
experiment to address the impact of eIF2 phosphorylation on arsenite treatment in our
reporter assay:

One day prior to reporter mRNA transfection, we transfected the cells with the plasmid
for overexpression of FLAG-tagged full length human *ppp1r15a* (GADD34)
phosphatase subunit. This almost completely abrogates eIF2 phosphorylation and the
inhibitory effect of arsenite on translation of reporters (see new panel D in Figure 4). Upon arsenite treatment, control Rluc
mRNA translation was downregulated 2 fold. Compare this with more than 6 fold when
GADD34 was overexpressed. Translation of *IFRD1* reporter was not
affected at all. From this experiment, we can conclude that inhibitory effect of
arsenite is mostly accounted for by eIF2 phosphorylation. This experiment is described
now in the penultimate paragraph of the subsection “5’ leaders of several
newly identified mRNAs are sufficient to provide resistance to the translation
inhibition”.

2) Regarding small disagreement between ribosome profiling experiments and reporter
assays:

We do not expect full concordance between these two assays. First, the signals provided
by these two approaches are quantitatively different. Ribosome profiling provides a
snapshot of translation after 0.5 hours of stress. The reporter assay relies on the
amount of luciferase produced during 2 hours post transfection. In other words, if we
think of translation as a function, ribosome profiling provides a signal at single time
point, while the reporter assay corresponds to the integral of the signal after a long
time period. Hence the only scenario when both approaches are expected to fully converge
would be when upon very rapid phosphorylation of eIF2 its levels would not change over
time. It is not the case with arsenite treatment when continuous increase of eIF2
phosphorylation is evident. Second, reporter assays do not depend on natural gene
expression levels, while ribosome profiling does. Because of that discordance is
particularly likely for low expressed genes. As noted by the referee,
*SLC35A4* is a good example of this. The change fold observed in
ribosome profiling data was high enough to surpass the threshold for statistical
significance (Z>4), yet the absolute level is unreliable due to high signal
variability for such a lowly expressed gene. Subsequently reporter assays confirmed
*SLC35A4* as the most resistant (even induced) mRNA, but did not
confirm the absolute levels.

3) Regarding the low levels of ATF4 mRNA preferential translation:

Assuming that all mRNAs identified in this study operate in eIF2-dependent manner, their
translation efficiency is likely to change during progression of the stress for the
reasons we outlined above. It is conceivable that some “preferentially
translated” mRNAs could become “resistant” when the level of active
eIF2 decreases further. Indeed, we observed increases in absolute levels of ATF4
reporter when a lower concentration of arsenite was used. Thus it is likely that the
concentration of arsenite is too high for the optimal levels of ATF4 translation. As we
pointed out earlier we did not aim to create conditions for the optimal ATF4
translation.

*Other comments*:

*1) The English is still a bit rough, but not to the point where the reader
cannot understand what is intended*.

We have made several minor alterations to the draft and have shown it to other
colleagues who are native English speakers.

*2) Although in the position of being a “break out” paper, it would
be nice if there was some effort expended at a possible explanation although it would
be obviously speculative (see below). To just leave the paper to end on “there
a black box that allows for this resistance to eIF2 repression” is
unsatisfying (although accurate)*.

Thank you. We conducted an experiment which may shed some light on the features required
for preferential/resistant translation. We observed earlier that all stress resistant
mRNAs possess efficiently translated uORFs. So, we hypothesized that some features of
5’ leaders upstream of uORFs may be important for the regulation. We decided to
address this issue with reporter constructs. To this end, we created two additional
reporters based on control pGL3 and *IFRD1*, where we added 5’
terminal stem loop of intermediate stability. As expected, addition of this stem loop
resulted in a 3-4 fold decrease of both reporters yield under normal conditions.
Interestingly, when arsenite stress was induced, SL+*IFRD1*
(*IFRD1* with stem loop) construct was no longer resistant, while
translation of SL+pGL3 and pGL3 constructs was similarly reduced. From this, we
can propose that high loading of preinitiator complexes to uORF is necessary for stress
resistance. Next, we addressed the question of whether the specific sequence upstream of
*IFRD1* uORF is required for the regulation. To this end, we
substituted it with an artificial single stranded (CAA)6 leader of the same length. This
modification did not affect stress resistance. Thus, we hypothesize that for resistant
translation, the uORF have to be situated under the control of a leader that allows high
initiation rates at uORF. These experiments are now described in a new subsection
“Unstructured leader sequence upstream of *IFRD1* uORF is
necessary for stress resistance”.

*3) The authors stress the resistance to eIF2 repression, but do not stress as
well that most of the proteins are poorly expressed. Using the number of reads scale
in*
Figure 2
*where ATF4 is 43, the other proteins fall mostly into the 0 to 2-8 range. This
inefficiency is also reflected in the reporter assays in*
Figure 4
*where pGL3 is 100%, the remaining normal configuration for most constructs
yields expression levels in the 5 to 10% of pGL3 range*.

Thank you. It is indeed so, as can be seen in Figure 3. We now mention this in the Discussion.

*4) That said, the low level of expression may be entirely appropriate for
phosphatases, kinases, transcription factors, etc. that are not required in large
amounts*.

We agree and thank you for providing a thought for the Discussion, which we have now
expanded to accommodate this suggestion.